# AIGCoder 1.0: Locally-Enhanced Language Modeling with Explicit and Structured Knowledge Memory

## Abstract

Large language models (LLMs) have achieved remarkable breakthroughs across various applications. However, their architectures remain inefficient due to two main limitations: (i) self-attention lacks an explicit inductive bias for locality, leading to redundant modeling of sequence-internal local information; (ii) mixture-of-experts (MoE) implicitly couples knowledge storage with computational pathways, hindering flexible access to sequence-external global knowledge. To overcome these limitations, we propose AIGCoder (AI Generative Coder), a novel LLM architecture that augments the standard decoder with two dedicated modules: 1) Local Fusion Attention (LFA), which incorporates a convolutional fusion to attention, explicitly capturing local patterns and allowing the attention to operate on more informative representations; 2) Knowledge Memory Module (KMM), which introduces a parametric key–value memory that explicitly stores global knowledge in addressable slots, decoupling storage from computation and enabling direct knowledge retrieval. Together, these modules enable AIGCoder to achieve more efficient and effective integration of information at both levels. Experimental results show that AIGCoder converges $1.33\times$ faster in pre-training than baseline models, underscoring its superiority over existing LLM architectures.

## 1 Introduction

Large language models (LLMs) (OpenAI, 2023; Touvron et al., 2023; Liu et al., 2024b) have achieved remarkable breakthroughs in natural language processing, demonstrating strong capabilities in applications such as customer service dialogue (Ou et al., 2024) and virtual assistants (Liu et al., 2025; Wang et al., 2025a). The core components of current LLMs primarily include 1) the attention mechanism (Vaswani et al., 2017), which models contextual dependencies by computing correlations across tokens in a sequence; and 2) the mixture of experts (MoE) (Shazeer et al., 2017), which enhances expressive power VIA expert routing and linear transformations. These components form the foundational architecture of modern LLMs, playing a central role for strong performance.

LLM architectures are expected to incorporate efficient mechanisms for modeling and processing information at different levels (Naveed et al., 2025): 1) Local-level information, which is internal to the input sequence, encompassing syntactic structures and short-range dependencies between adjacent tokens. Modeling such local structures is essential for interpreting the immediate semantic context within a sequence (Yang et al., 2021; Chen et al., 2025). 2) Global-level information, conversely, which is independent of any specific input sequence, refers to the generic knowledge (*e.g.*, commonsense, factual knowledge). This can be internalized during pre-training, forming a reservoir essential for deep reasoning (Geva et al., 2021; Mu & Lin, 2025). The challenge lies in effectively combining these local patterns with the global knowledge to empower LLMs for complex tasks.

However, we argue that existing LLM architectures are computationally inefficient in integrating these two levels of information (See empirical analysis in Section 5.4). For modeling sequence-internal local information, while attention mechanisms can capture both short- and long-range dependencies, it does so through redundant full pairwise interactions, which makes learning local patterns computationally inefficient. For incorporating sequence-external global information, MoE models implicitly distribute knowledge across the weights of expert networks. This approach cou-

ples knowledge storage with computational transformation, making knowledge access indirect and inflexible for direct retrieval or targeted updates. Thus, designing an architecture that efficiently and effectively integrates local and global information remains an open challenge.

In this paper, we propose AIGCoder (AI Generative Coder), a novel LLM architecture built upon an enhanced decoder block. It extends the vanilla decoder with two new modules: the Local Fusion Attention (LFA) and the Knowledge Memory Module (KMM), which are designed to handle sequence-internal local information and sequence-external global information. For modeling local information, LFA incorporates a convolution operation that fuses adjacent token representations before the attention mechanism. This design introduces an explicit local inductive bias, allowing the model to capture short-range dependencies more efficiently while enabling the attention layer to concentrate on modeling broader contextual relationships. For incorporating global knowledge, KMM introduces a parameterized key–value memory structure that explicitly stores knowledge in addressable slots. This setup decouples knowledge storage from computational pathways, enabling tokens to directly query and retrieve relevant knowledge, thereby enhancing the transparency and flexibility of knowledge access. By integrating LFA and KMM, AIGCoder achieves a more effective and efficient fusion of information at both levels, leading to improved representational capacity and adaptability in tackling complex NLP tasks. Our novelty and main contributions are as follows:

- We identify the inefficiency of self-attention for short-range dependencies modeling. To address this, our LFA decouples the modeling process by employing a convolutional fusion layer to explicitly capture local patterns before attention. This inductive bias relieves the attention mechanism from redundantly modeling adjacent tokens, enabling it to concentrate on modeling more complex and global contextual relationships.

- We argue that the implicit knowledge representation in MoE couples knowledge storage with computational pathways, limiting transparent access and flexible manipulation. We propose KMM, a parametric key-value memory that explicitly stores global knowledge in addressable slots. This design decouples storage from computation, enabling direct knowledge retrieval, potentially easier knowledge editing, and more interpretable inference.

- Our AIGCoder achieves a $1.33\times$ faster convergence than baseline in pretraining, translating directly into substantial savings in computational costs and energy consumption. It attains the same loss level on the validation set in only 7.5K training steps that the baseline requires 10K steps to reach, demonstrating the superior parameter efficiency and optimization effectiveness brought by our architectural innovations.

## 2 RELATED WORK

**Efficient Attention Mechanisms**. Recent advances aim to mitigate the quadratic complexity of standard self-attention by approximating full attention through sparse token interactions. Static approaches (Child et al., 2019) employ predefined and input-agnostic patterns, such as local windows (Beltagy et al., 2020), random connections (Zaheer et al., 2020), and global tokens (Xiao et al., 2024). Dynamic sparse attention (Jiang et al., 2024b) adapts the attention pattern to the input content, enabling context-aware computation through token-level pruning (Ren et al., 2023), heavy-hitter selection (Zhang et al., 2023), dynamic group aggregation (Zhang et al., 2025), or global-local fusion (Chen et al., 2025). Despite their effectiveness in lowering FLOPs or memory usage, these methods primarily aim at global computation reduction. In contrast, our Local Fusion Attention does not reduce sequence-level complexity but introduces an explicit local inductive bias, enabling more efficient modeling of short-range dependencies without altering the full attention structure.

**Mixture-of-Experts**. The MoE framework enables scalable language modeling by activating only a subset of experts per token, thus increasing model capacity without proportional computational cost (Shazeer et al., 2017; Fedus et al., 2022). Recent work improves efficiency through better routing strategies, such as load-balanced gating (Fedus et al., 2022), expert choice mechanisms (Zhou et al., 2022), and stable training formulations (Dai et al., 2022; Pan et al., 2024). Model initialization has also been simplified by constructing experts from dense checkpoints (Komatsuzaki et al., 2023; Wei et al., 2024). Despite these advances, knowledge in MoE remains implicitly encoded within expert parameters and can only be accessed via computation. In contrast, our method introduces an explicit, parameterized key-value store that supports direct querying and reusable knowledge.

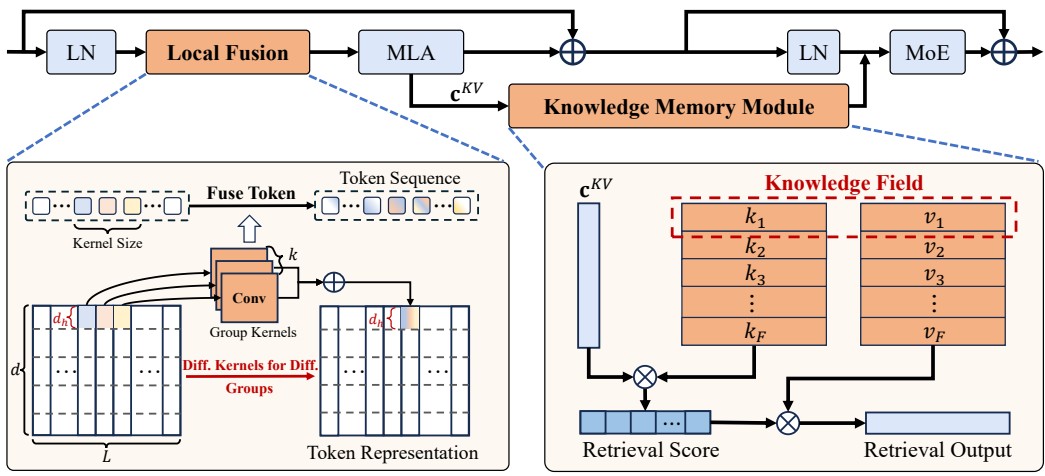

Figure 1: Overall architecture of the decoder layer in our AIGCoder. The model enhances a vanilla decoder block with two proposed modules: Local Fusion Attention (LFA) and the Knowledge Memory Module (KMM). Together with the Multi-Head Latent Attention (MLA) backbone and Mixture-of-Experts (MoE) layer, these components provide complementary local and global modeling capabilities, improving both modeling efficiency and knowledge exploitation.

## 3 PRELIMINARY: MULTI-HEAD LATENT ATTENTION

Recently, DeepSeek (Liu et al., 2024b) introduces multi-head latent attention (MLA) mechanism, which compresses contextual representations into a lower-dimensional latent space to improve the efficiency of long-context modeling. Given an input sequence $\mathbf{U}=\{\mathbf{u}_1, \mathbf{u}_2, \ldots, \mathbf{u}_L\} \in \mathbb{R}^{L \times d}$ of $L$ tokens with dimension $d$, MLA first projects each token into a compact latent representation:

$$
\begin{aligned}
\mathbf{c}^{\mathrm{Q}} &= \mathbf{U}\mathbf{W}^{\mathrm{QD}}, \quad \mathbf{c}^{\mathrm{Q}} \in \mathbb{R}^{L \times d'_c}, \quad d'_c < d, \\
\mathbf{c}^{\mathrm{KV}} &= \mathbf{U}\mathbf{W}^{\mathrm{KVD}}, \quad \mathbf{c}^{\mathrm{KV}} \in \mathbb{R}^{L \times d_c}, \quad d_c < d,
\end{aligned}
\tag{1}
$$

where $\mathbf{W}^{\mathrm{QD}} \in \mathbb{R}^{d \times d'_c}$ and $\mathbf{W}^{\mathrm{KVD}} \in \mathbb{R}^{d \times d_c}$ are the down-projection matrices. The latent representation is then expanded to obtain query, key, and value through $h$ parallel linear projection. Finally, the multi-head latent attention is computed as via $h$ parallel attention heads:

$$
\begin{aligned}
\mathrm{MLA}(\mathbf{U}) &= [\mathbf{Att}^1, \ldots, \mathbf{Att}^h] \in \mathbb{R}^{L \times d}, \\
\mathbf{Att}^i &= \mathrm{softmax}\left(\frac{\mathbf{Q}^i \mathbf{K}^{i\top}}{\sqrt{d_h}}\right) \mathbf{V}^i, \\
\mathbf{Q}^i &= \mathbf{c}^{\mathrm{Q}}\mathbf{W}^{Q_i}, \ \mathbf{K}^i = \mathbf{c}^{\mathrm{KV}}\mathbf{W}^{K_i}, \ \mathbf{V}^i = \mathbf{c}^{\mathrm{KV}}\mathbf{W}^{V_i},
\end{aligned}
\tag{2}
$$

with $\mathbf{W}^{Q_i} \in \mathbb{R}^{d'_c \times d_h}, \mathbf{W}^{K_i} \in \mathbb{R}^{d_c \times d_h}, \mathbf{W}^{V_i} \in \mathbb{R}^{d_c \times d_h}$ as learnable parameters, $d_h$ denotes the token dimension in each head, $i$ denotes the index of heads. For clarity, we omit the details of rotary position embeddings (RoPE) (Su et al., 2024) in the above formulation. In practice, our implementation follows the same RoPE integration as in the original DeepSeek work (Liu et al., 2024b), and this simplification is made solely for ease of presentation. It is worth noting that, similar to vanilla self-attention, MLA still models short- and long-range dependencies in a undifferentiated manner, which may limit its efficiency in capturing fine-grained local patterns.

## 4 AIGCODER: INDUCTIVE BIAS AND KNOWLEDGE MEMORY FOR LLM

### 4.1 METHOD OVERVIEW

In this paper, we propose **AIGCoder** (AI Generative Coder), a novel LLM architecture that enhances the vanilla decoder block with two complementary modules: Local Fusion Attention (LFA)

---

**Algorithm 1** Computation Flow of Decoder Block in our AIGCoder.

---

**Require:** Hidden states $\mathbf{U} = \{\mathbf{u}_1, \mathbf{u}_2, \ldots, \mathbf{u}_L\} \in \mathbb{R}^{L \times d}$; knowledge field matrices $\mathcal{K}$ and $\mathcal{V}$; learnable parameters $\mathbf{W}^{QD}, \mathbf{W}^{KVD}, \mathbf{W}^Q, \mathbf{W}^K, \mathbf{W}^V, \mathbf{W}^H, \mathbf{W}^O$; hyperparameters $h$, $k$ and $c$.

1: Compute locally fused representation $\hat{\mathbf{U}} = \text{Conv}(\mathbf{U})$ via Eqn. (3) with kernel size $k$ and group count $h$.
2: Compute latent representations: $\mathbf{c}^Q = \hat{\mathbf{U}}\mathbf{W}^{QD} \in \mathbb{R}^{L \times d'_c}$ and $\mathbf{c}^{KV} = \hat{\mathbf{U}}\mathbf{W}^{KVD} \in \mathbb{R}^{L \times d_c}$.
3: Compute multi-head latent attention: $\mathbf{O}^A = \text{MLA}(\mathbf{c}^Q, \mathbf{c}^{KV}; \mathbf{W}^Q, \mathbf{W}^K, \mathbf{W}^V)$ via Eqn. (2) with $h$ heads.
4: Execute knowledge memory module: $\mathbf{H} = \mathbf{c}^{KV}\mathbf{W}^H$, $\mathbf{Z} = \text{KMM}(\mathbf{H}; \mathcal{K}, \mathcal{V})$ via Eqn. (6), $\mathbf{O}^K = \mathbf{Z}\mathbf{W}^O$.
5: Compute expert-enhanced output: $\mathbf{O} = \text{MoE}(\mathbf{O}^A + \mathbf{O}^K)$.
**Ensure:** Final block output $\mathbf{O}$.

---

and Knowledge Memory Module (KMM). These modules together improve the effectiveness and efficiency of information exploitation at both local and global levels. Our LFA extends the Multi-Head Latent Attention (MLA) (Liu et al., 2024b) framework by introducing an explicit local inductive bias (c.f. Section 4.2). Instead of relying solely on pairwise interactions, LFA applies convolutional fusion to adjacent tokens before attention. This design enables more efficient capture of short-range dependencies while providing the subsequent attention mechanism with richer contextual representations for modeling broader contextual relationships.

At the global level, our KMM provides an explicit key–value memory for knowledge access. It takes the latent representation $\mathbf{c}$ as input, projects it into queries, matches them against learnable keys that encode commonsense and domain-specific knowledge fields. Subsequently, we aggregate the corresponding values to produce the output (c.f. Section 4.3). Finally, the output of KMM is combined with the MoE result to form the final decoder-block representation. In this case, KMM augments the implicit parameterization of MoEs with an explicit, reusable memory pathway, improving transparency and flexibility in knowledge access. The overview and pseudo code of our proposed architecture are shown in Figure 1 and Algorithm 1, respetively.

## 4.2 LOCAL FUSION ATTENTION FOR EFFICIENT SHORT DEPENDENCY MODELING

Natural language exhibits strong locality, where adjacent tokens are highly correlated at the semantic levels. Prior works [xx] have demonstrated that local modeling improves representation performance. However, both vanilla self-attention (Vaswani et al., 2017) and multi-head latent attention (MLA) (Liu et al., 2024b) model short-range dependencies together with long-range ones, requiring global pairwise interactions even when only local patterns are needed. This may be inefficient for capturing local fine-grained relationships between adjacent tokens. To alleviate this issue, we propose **Local Fusion Attention (LFA)**, which enhances the attention by introducing a group convolution that explicitly fuses features of neighboring tokens (validated in ablation Figure 4). This design provides a strong local inductive bias, enabling more efficient modeling of short-range dependencies while preparing better contextualized representations.

Formally, we split the hidden states $\mathbf{U} \in \mathbb{R}^{L \times d}$ along the feature dimension into $h$ groups, each of size $d_h = d/h$: $\mathbf{U} = [\mathbf{U}^{(1)}, \mathbf{U}^{(2)}, \cdots, \mathbf{U}^{(h)}]$, where $\mathbf{U}^{(g)} \in \mathbb{R}^{L \times d_h}$ denotes the hidden states for the group $g$. The number of groups is set equal to the number of attention heads $h$, so that each head receives its own locally fused representation and learn distinct local fusion patterns in parallel. For each group $g \in [1, h]$, we apply a 1D convolution with a kernel $\Theta^{(g)} \in \mathbb{R}^{k \times d_h \times d_h}$ along the sequence dimension to aggregate information from its $k$ immediate neighbors within the same group, creating a richer, context-aware representation before being projected into the latent space for attention. The locally fused representation for group $g$ is computed as:

$$\hat{\mathbf{U}}_t^{(g)} = \sum_{s=0}^{k-1} \mathbf{U}_{t-s+\lfloor k/2 \rfloor}^{(g)} \cdot \Theta^{(g)}[s], \tag{3}$$

where $\Theta^{(g)}[s] \in \mathbb{R}^{d_h \times d_h}$ is the slice of the kernel at position $s$. the index $t - s + \lfloor k/2 \rfloor$ explicitly moves along the sequence axis $(L)$, meaning that each output position aggregates information from its $k$ neighboring tokens within the same group. Finally, we concatenate all groups to obtain $\hat{\mathbf{U}} = [\hat{\mathbf{U}}^{(1)}, \hat{\mathbf{U}}^{(2)}, \cdots, \hat{\mathbf{U}}^{(h)}]$. Note that we use a stride of 1 and apply suitable padding so that the sequence length remains unchanged, *i.e.*, $\hat{\mathbf{U}}^{(g)} \in \mathbb{R}^{L \times d_h}$. The locally fused representation $\hat{\mathbf{U}}$ is

then used for query, key and value construction via Eqn.(1). Our proposed LFA is a lightweight modification introduces, only introducing a small number of additional parameters (only 0.41% in our AIGCoder-7B model). Empirical results show that LFA accelerates convergence in training (c.f. Figure 4) and consistently enhances downstream performance (c.f. Tables 1 and 2), demonstrating its effectiveness despite the minimal overhead.

## 4.3 Knowledge Memory Module for Explicit Knowledge Fields Modeling

While the proposed LFA focuses on enhancing the modeling of short-range dependencies, effectively leveraging global-level knowledge remains a challenge. Existing approaches such as MoE layers (Shazeer et al., 2017) implicitly encode knowledge within distributed parameters, which must be re-activated through linear transformation for every query. This implicit storage lacks transparency and restricts efficient reuse of knowledge patterns. To address this, we propose a **Knowledge Memory Module (KMM)**, which explicitly organizes global knowledge into parameterized key–value fields that can be directly queried during model execution (validated in ablation Figure 4).

Instead of directly using the original hidden states $\mathbf{U}$, KMM takes the latent representation $\mathbf{c}^{\text{KV}} \in \mathbb{R}^{L \times d_c}$ from MLA as input. This compact representation is both computationally efficient and semantically rich, making it well-suited for querying global knowledge fields. We first project it to query space with learnable parameters $\mathbf{W}^{\text{H}} \in \mathbb{R}^{d_c \times d_u}$:

$$\mathbf{H} = \mathbf{c}^{\text{KV}} \mathbf{W}^{\text{H}}, \tag{4}$$

where $\mathbf{H} \in \mathbb{R}^{L \times d_u}$ denotes the query features projected from the latent KV representation $\mathbf{c}^{KV}$.

**Multi-group knowledge retrieval over parameterized fields.** KMM adopts a multi-group retrieval scheme to increase capacity and capture heterogeneous semantics. Formally, we store the knowledge fields as parameterized keys and values:

$$\mathcal{K} \in \mathbb{R}^{F \times d_u}, \quad \mathcal{V} \in \mathbb{R}^{F \times d_v}, \tag{5}$$

where $F$ denotes the number of fields, $d_u$ is the query/key dimension, and $d_v$ is the value dimension. Note that $\mathcal{K}$ and $\mathcal{V}$ are randomly initialized and optimized end-to-end during training. We split the projected query $\mathbf{H}$ and the knowledge field parameters $\mathcal{K}, \mathcal{V}$ along the feature dimension into $c$ groups, where each group corresponds to a lower-dimensional subspace. In each group, KMM computes the similarity between the group-specific query $\mathbf{H}^{(i)}$ and the knowledge keys $\mathcal{K}^{(i)}$ using scaled dot-product, and then aggregates the corresponding values $\mathcal{V}^{(i)}$ to obtain the retrieved representation $\mathbf{Z}^{(i)}$. The outputs from all groups are finally concatenated to form the overall representation:

$$\text{KMM}(\mathbf{H}; \mathcal{K}, \mathcal{V}) = [\mathbf{Z}^{(1)}, \ldots, \mathbf{Z}^{(c)}] \in \mathbb{R}^{L \times d},$$

$$\mathbf{Z}^{(i)} = \text{softmax}\left(\frac{\mathbf{H}^{(i)} \mathcal{K}^{(i)\top}}{\sqrt{d_u}}\right) \mathcal{V}^{(i)}. \tag{6}$$

The knowledge-enhanced output in Eqn. (6) then will be projected back to the model dimension through a linear transformation with learnable parameters $\mathbf{W}^{\text{O}} \in \mathbb{R}^{d_v \times d}$, yielding $\mathbf{O}^{\text{K}}$. This representation is then combined with the output of the MLA, denoted as $\mathbf{O}^{\text{A}}$, by simple addition: $\mathbf{O}^{\text{A}} + \mathbf{O}^{\text{K}}$. The combined representation is subsequently processed by MoE layer[1], producing the final output: $\mathbf{O} = \text{MOE}(\mathbf{O}^{\text{A}} + \mathbf{O}^{\text{K}})$. In this way, the final output integrates the generative expressiveness of MoE with the explicit knowledge reuse enabled by KMM, allowing the model to balance the creation of novel patterns with the efficient retrieval of knowledge fields.

**Differences of KMM from Attention**. The roles and design principles of KMM and attention are fundamentally different. In attention, both keys and values are dynamically computed from the input, making it primarily responsible for modeling contextual dependencies. In contrast, KMM employs fixed and parameterized keys and values during inference and serve as explicit knowledge fields. This makes KMM closer to a reusable memory, complementary to MoE: while MoE generates new patterns through expert routing, KMM provides efficient retrieval of explicit knowledge.

**Visualization of Knowledge Field-Domain Associations**. To provide an initial validation of our design, we conduct a qualitative analysis by examining the model's behavior on five diverse domains in MMLU (Hendrycks et al., 2021) benchmark. In Figure 2, the results demonstrates that our

---

[1]Details of the employed MoE architecture are provided in the supplementary material.

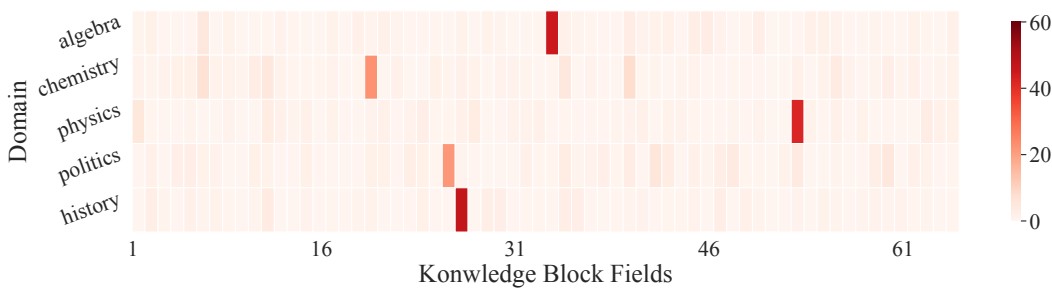

Figure 2: Field–domain associations of the proposed KMM with 64 knowledge fields, evaluated on five MMLU domains (1,024 samples each). The heatmap of softmax($\mathbf{H}\mathcal{K}/d_u$) in layer 4 shows that fields emerge with domain-specific specialization, enabling explicit and interpretable knowledge retrieval. We put more visualizations of the remaining layers in Section E of the supplementary.

KMM facilitates the emergence of semantically specialized knowledge fields. The analysis reveals that, through end-to-end training, individual fields spontaneously evolve to represent concepts from specific domains (*e.g.*, Field 25 is consistently activated for history-related questions, while Field 51 is specialized for physics). The specialization is statistically robust, as evidenced by consistent activation patterns aggregated across multiple layers and a large number of samples. This observed self-organization into a structured memory is a key advantage of our explicit design over implicit approaches like MoE. This enables targeted knowledge retrieval, much like consulting an expert. This makes the model's predictions more interpretable and reliable, as the knowledge source is directly observable.

## 5 EXPERIMENTS

### 5.1 EXPERIMENTAL SETUP

**Models**. Based on the proposed AIGCoder architecture, we constructed a series of models with different parameter scales by adjusting the number of layers and the hidden state dimension. Specifically, we trained models of sizes 1B, 5B, 7B, 13B, 33B, and 60B parameters. This scaling setup allows us to systematically examine the effectiveness and stability of AIGCoder across a wide range of capacities. In the main experiments, we focus on comparisons of our AIGCoder-7B. We provide detailed model specifications in Section C of the supplementary.

**Dataset**. We pre-train our model on the publicly released Matrix Data Pile, a comprehensive bilingual corpus of 4.5 trillion high-quality tokens curated for the MAP-Neo series (Zhang et al., 2024), comprising re-processed high-quality English datasets (e.g., RedPajama (Weber et al., 2024), Dolma (Soldaini et al., 2024)) and Chinese datasets (e.g., Skypile (Wei et al., 2023), ChineseWebText (Chen et al., 2023)), along with a large volume of newly crawled Chinese web content. For validation, we randomly sample 1% from the English Common Crawl (cc_en) portions of the Matrix Data Pile to form validation sets of approximately 18B, ensuring representativeness and efficiency. For supervised fine-tuning (SFT), we employ a curated dataset of approximately 10 billion high-quality tokens, collected and filtered from existing public instruction-following corpora. More details can be found in Sections D.1 and D.2 in the supplementary.

**Metric**. We evaluate our model across four core dimensions using standard benchmarks. For language understanding, we report results on MMLU (Hendrycks et al., 2021) for English, and CMMLU (Li et al., 2024), C-Eval (Huang et al., 2023) for Chinese. Reasoning is assessed with HellaSwag (Zellers et al., 2019) and ARC-Challenge (Clark et al., 2018). We measure code generation performance via pass@1 on HumanEval (Chen et al., 2021), and evaluate mathematical reasoning using GSM8K (Cobbe et al., 2021). We put more details in Section D.1 in the supplementary.

**Baseline Models**. We compare AIGCoder-7B against three categories of baselines: (i) size-comparable models, including Llama-3.1-8B (Grattafiori et al., 2024), Qwen2.5-7B (Yang et al., 2024), Gemma-7B (Team et al., 2024b), InternLM2-7B (Cai et al., 2024), Phi-3-medium (Abdin et al., 2024), Mixtral-8×7B (Jiang et al., 2024a), and DeepSeek-MoE-16B (Dai et al., 2024); (ii)

Table 1: Comparisons of AIGCoder-7B with **size-comparable LLMs** across language, reasoning, code, math domains. **Bold** and underlined numbers indicate the best and second-best results, respectively. "HellS." and "HumE." are short for "HellaSwag" and "HumanEval", respectively.

| Model | Arch. | # Act. | # Total | Language | | | Reasoning | | Code | Math |
|---|---|---|---|---|---|---|---|---|---|---|
| | | | | MMLU | CMMLU | C-Eval | HellS. | ARC-C | HumE. | GSM8K |
| Llama-3.1-8B | Dense | 8B | 8B | 73.0 | – | 52.0 | 82.3 | 83.4 | 72.6 | 84.5 |
| Qwen2.5-7B | Dense | 7B | 7B | 76.6 | 79.1 | 76.2 | 81.5 | 63.4 | 84.8 | **91.6** |
| Gemma-7B | Dense | 7B | 7B | 64.3 | – | – | 81.2 | 53.2 | 32.3 | 46.4 |
| InternLM2-7B | Dense | 7B | 7B | 63.7 | 63.0 | 60.8 | 83.0 | – | 59.2 | 70.7 |
| Phi-3 medium | Dense | 3.8B | 3.8B | 78.0 | – | – | 82.4 | 91.6 | 62.2 | 91.0 |
| Mixtral-8×7B | MoE | 14B | 46.7B | 70.5 | – | – | 70.4 | 87.3 | 37.8 | 64.7 |
| DeepseekMoE-16B | MoE | 2.4B | 16B | 47.2 | 49.3 | 40.0 | 72.2 | 50.0 | 45.7 | 62.2 |
| AIGCoder-7B (Ours) | MoE | 3B | 7B | **91.5** | **93.4** | **92.8** | **89.5** | **96.2** | **87.0** | 73.0 |

frontier open-source models, including Llama-3.1-70B, Llama-3.1-405B (Grattafiori et al., 2024), Qwen2.5-72B (Yang et al., 2024), Mixtral-8×22B (Jiang et al., 2024a), DeepSeek-V2.5 (Liu et al., 2024a), and Hunyuan-Large (Sun et al., 2024); and (iii) frontier closed-source models, including Claude-3.5-Sonnet-1022 (Anthropic, 2024) and Gemini-1.5-Pro (May'24) (Team et al., 2024a). All evaluations are performed using their instructed versions.

**Implementation Details**. We build models of different scales based on the architecture illustrated in Fig. 1 and train them with the Megatron-LM framework on clusters equipped with H200 or Ascend 910B GPUs. For the ablations (c.f. Section 5.4), we use 5B model to investigate the effectiveness of key components. In this setting, we employ a global batch size of 1,024 and a context length of 2048. We train the model for 10k steps, consuming approximately 21B tokens. For the main pre-training runs, the 7B and 13B models use a global batch size of 16,384 and a 2048 context for 134k steps (about 4.5T tokens), while the 33B and 60B models employ a 4096 context with the same total tokens, resulting in roughly half the number of steps (67k). Specific hyperparameters are tuned based on model scale and hardware (see Section D.3). The 7B model used in our evaluations undergoes a two-stage training process. After pre-training, it is fine-tuned via supervised fine-tuning (SFT) with reinforcement learning process on a curated dataset of 10B high-quality instruction-following tokens (see Section D.2 for details), ensuring no leakage from downstream evaluation sets.

## 5.2 Performance Comparisons

**Comparisons with Size-comparable Models**. We first compare our AIGCoder-7B with size-comparable open-source models. We report the results across four domains in Table 1. Despite using a similar parameter scale, our AIGCoder achieves substantial improvements. For instance, in language benchmarks, it reaches 91.5 on MMLU, 93.4 on CMMLU, and 92.8 on C-Eval, surpassing all dense and MoE baselines by large margins. In reasoning tasks, AIGCoder attains 89.5 on HellaSwag and 96.2 on ARC-C, again outperforming the best size-comparable models. These improvements are largely attributed to the proposed LFA and KMM modules, which enhance local dependency modeling and explicit knowledge utilization. Overall, these results show AIGCoder delivers state-of-the-art performance among size-comparable LLMs.

**Comparisons with Frontier Open-source Models**. We further compare AIGCoder-7B against leading open-source LLMs with much larger parameter scales, including dense models. Results are summarized in Table 2. Despite using only 7B total parameters with 3B activated, AIGCoder consistently surpasses these frontier systems across most domains. For instance, in language understanding, it achieves 91.5 on MMLU, 93.4 on CMMLU, and 92.8 on C-Eval, far exceeding the best dense and MoE baselines. In code generation, AIGCoder achieves 87.0 on HumanEval, roughly comparable to Hunyuan-Large (90.0). These results highlight the efficiency and effectiveness of our proposed AIGCoder: with less than one-tenth of the activated parameters, it delivers SoTA performance across most domains, underscoring the advantages of our LFA and KMM modules.

**Comparisons with Frontier Closed-source Models**. Besides, we compare AIGCoder-7B with frontier closed-source systems, including Claude-3.5-Sonnet and Gemini-1.5-Pro, as summarized in Table 3. Despite its much smaller scale, AIGCoder demonstrates highly competitive performance. In

Table 2: Comparisons of AIGCoder-7B with **frontier open-source LLMs** across language, reasoning, code, math domains. **Bold** and underlined numbers indicate the best and second-best results, respectively. "HellS." and "HumE." are short for "HellaSwag" and "HumanEval", respectively.

| Model | Arch. | # Act. | # Total | Language | | | Reasoning | | Code | Math |
|---|---|---|---|---|---|---|---|---|---|---|
| | | | | MMLU | CMMLU | C-Eval | HellS. | ARC-C | HumE. | GSM8K |
| Llama-3.1-70B | Dense | 70B | 70B | 83.6 | 69.0 | – | 86.7 | 94.8 | 80.5 | 95.1 |
| Llama-3.1-405B | Dense | 405B | 405B | 86.0 | – | 61.5 | 88.3 | **96.9** | 89.0 | **96.8** |
| Qwen2.5-72B | Dense | 72B | 72B | 84.4 | 86.7 | 84.7 | – | – | 86.6 | 95.8 |
| Mixtral-8×22B | MoE | 39B | 141B | 77.8 | 61.0 | 60.0 | 89.2 | 90.0 | 75.0 | 85.0 |
| Deepseek-V2.5 | MoE | 21B | 236B | 80.4 | – | – | 85.0 | 72.6 | 89.0 | 88.3 |
| Hunyuan-Large | MoE | 52B | 389B | 89.9 | 90.4 | 89.5 | – | 94.6 | **90.0** | – |
| AIGCoder-7B (Ours) | MoE | 3B | 7B | **91.5** | **93.4** | **92.8** | **89.5** | 96.2 | 87.0 | 73.0 |

Table 3: Comparisons of AIGCoder-7B with **frontier closed-source LLMs** across language, reasoning, code, math domains. **Bold** and underlined numbers indicate the best and second-best results, respectively. "HellS." and "HumE." are short for "HellaSwag" and "HumanEval", respectively.

| Model | MMLU | HellS. | HumE. | GSM8K |
|---|---|---|---|---|
| Claude-3.5-Sonnet-1022 | 88.3 | **94.6** | **92.0** | **96.4** |
| Gemini-1.5-Pro (May'24) | 85.9 | 93.3 | 84.1 | 90.8 |
| AIGCoder-7B (Ours) | **91.5** | 89.5 | 87.0 | 73.0 |

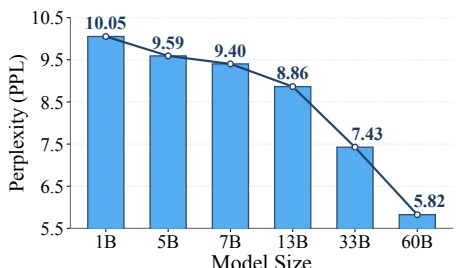

Figure 3: Validation PPL on WikiText-103 across models from 1B to 60B parameters.

language understanding, it achieves 91.5 on MMLU, outperforming both Claude-3.5-Sonnet (88.3) and Gemini-1.5-Pro (85.9) by large margins. These results highlight that AIGCoder can deliver competitive or superior performance to frontier closed-source systems on several challenging benchmarks, despite operating at a fraction of their parameter scale.

## 5.3 DEMONSTRATION OF SCALABILITY

We evaluated the scalability of AIGCoder across model sizes from 1B to 60B parameters. As shown in Figure 3, PPL decreases consistently with larger capacity, from 10.05 at 1B to 5.82 at 60B, confirming smooth scaling behavior. In addition, the results indicates that the proposed components, LFA and KMM, remain effective across scales. Intermediate models also follow this trend (9.59 at 5B, 9.40 at 7B, 8.86 at 13B, and 7.43 at 33B), suggesting predictable improvements as model size grows. Training further exhibited stable optimization dynamics, with maximum gradient norms remaining below 1.0 across all models and no collapse or divergence observed, demonstrating that AIGCoder can be scaled reliably to tens of billions of parameters.

## 5.4 FURTHER EXPERIMENTS

We conduct all ablation studies on AIGCoder-5B, which is sufficiently lightweight to allow systematic experimentation while still large enough to yield representative results.

**Effect of Proposed Components on Convergence Speed**. We investigate the effect of different components by progressively add our proposed modules in the baseline. In Figure 4, at 10k training steps, the PPL of the baseline, baseline+LFA, and baseline+LFA+KMM (*i.e.*, our AIGCoder) are 31.82, 30.57, and 28.50 on the training set, and 31.82, 30.88, and 29.08 on the evaluation set, respectively. Using the baseline's 10k-step PPL (31.82) as a reference, the LFA-augmented model reaches this level at 9k steps, showing 1.11× faster convergence. With both LFA and KMM, our AIGCoder achieves the same target at 7.8k (train) and 7.5k (eval) steps, corresponding to 1.28× and 1.33× faster convergence. These show that LFA and KMM improves convergence efficiency.

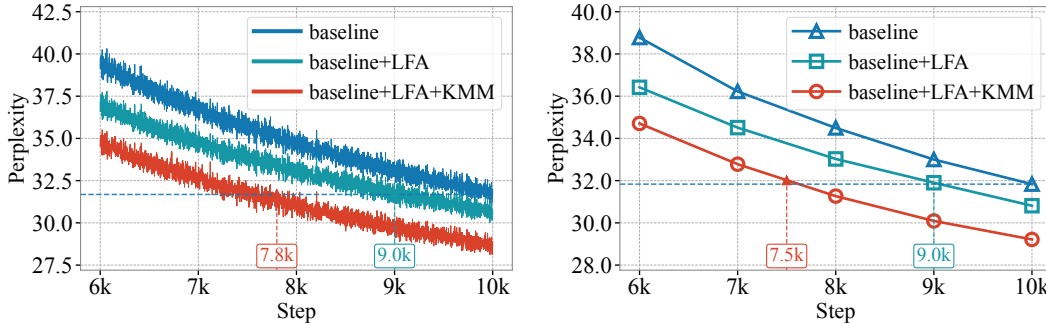

Figure 4: Ablation study of the proposed components LFA and KMM. Left and right show training loss curves and evaluation loss on the validation set at fixed steps, respectively. All variants are pretrained for 10k steps under the same settings, and performance is measured by loss.

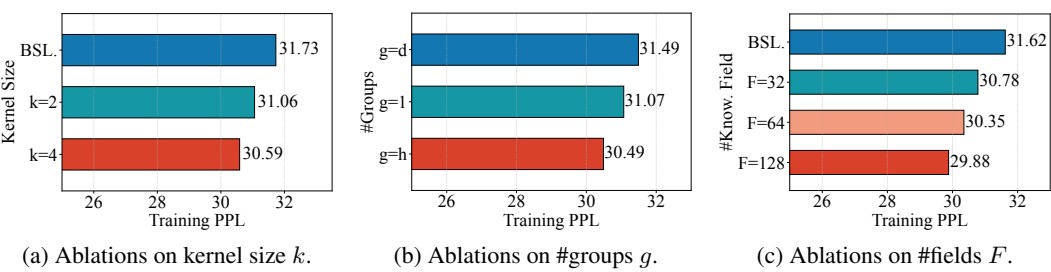

(a) Ablations on kernel size $k$.      (b) Ablations on #groups $g$.      (c) Ablations on #fields $F$.

Figure 5: Ablations on the kernel size $k$ and the number of convolutional group $g$ in LFA, and the number of knowledge fields $F$ in KMM. "BSL." is short for "baseline".

**Ablations of Kernel Sizes $k$ in Local Fusion**. We ablate the kernel size $k$ of the convolution in the local fusion module with different $k$ (*i.e.*, $k = 2$ and $k = 4$). Under identical training settings, we report training PPL at the 10k training step. In Figure 5 (a), training PPL decreases from 31.73 (baseline) to 31.06 ($k = 2$) and 30.59 ($k = 4$), respectively. Within the tested range, we adopt $k = 4$ as a practical balance between effectiveness and efficiency. It performs better than $k = 2$ in our ablations, and due to the high computational cost of large-scale pre-training, we did not explore larger values. The chosen setting already provides strong performance gains with manageable overhead.

**Ablations of Number of convolutional groups $g$ in Local Fusion**. We vary the number of groups $g$ in the convolution of the LFA module, comparing $g = 1$ (no grouping), $g = d$ (grouped by token feature dimension), and $g = h$ (aligned with the number of attention heads). Under identical training settings, we report training PPL at the 10k step. In Figure 5 (b), training PPL is 31.07 for $g = 1$, 31.49 for $g = d$, and 30.49 for $g = h$. Using $g = 1$ as the reference, $g = h$ reduces PPL by 1.87%, whereas $g = d$ slightly worsens it by 1.35%; relative to $g = d$, $g = h$ achieves a 3.18% lower PPL. These findings suggest that grouping in LFA module is necessary, and that the group count should be aligned with the attention heads for the best performance.

**Ablations of Number in Knowledge Fields $F$**. We conduct ablations on the number of knowledge fields $F$ in KMM. In Figure 5 (c), training PPL on cc-en dataset decreases monotonically as $F$ increases: from 31.62 (baseline without KMM) to 30.78 ($F$=32), 30.35 ($F$=64), and 29.88 ($F$=128). These correspond to relative reductions of 2.66%, 4.02%, and 5.50% over the baseline, respectively. While $F$=128 yields the lowest PPL, $F$=64 already captures about 73% of the total reduction achieved at $F$=128, with substantially lower computational and memory cost. Hence, we adopt $F$=64 as a balanced configuration between efficiency and performance in the experiments.

## 6 CONCLUSION AND FUTURE WORK

In this work, we propose AIGCoder, a novel large language model architecture that enhances the vanilla Transformer decoder block with two complementary modules: Local Fusion Attention (LFA) for efficient short-range dependency modeling and the Knowledge Memory Module (KMM) for explicit knowledge retrieval. These modules together improve both local contextual modeling and global knowledge utilization, leading to faster convergence and stronger performance.

**Future Work: Multimodal Extension**. A promising future direction is extending AIGCoder to multimodal reasoning. Its standard Transformer backbone naturally supports unified sequences of tokenized embeddings (*e.g.*, from text, images, or audio), requiring no architectural changes. We plan to explore how its LFA and KMM can enhance multimodal understanding and generation.

## REPRODUCIBILITY STATEMENT

In this work, we pretrain AIGCoder models of varying sizes on the Matrix Data Pile dataset. Reproducing all the results presented in our study depends on the following three aspects:

1. **DATASET.** We provide a comprehensive description of the dataset adopted in this study in the second paragraph of Section 5.1 as well as in Appendix D.

2. **IMPLEMENTATION DETAILS.** Comprehensive implementation details are elaborated in the fifth paragraph of Section 5.1 and Appendix D. We will open-source the codebase, training scripts, and the pre-trained models upon publication.

3. **COMPUTE AND LIBRARIES.** We train our AIGCoder models on GPU clusters. For model scales up to and including 13B parameters, we use a cluster of 20 nodes, each equipped with 8 Ascend 910B GPUs (64GB memory each). The software stack is built on Ascend PyTorch 6.0-RC3. For larger-scale models (*e.g.*, 33B or 60B), training is scaled to 50 nodes (8 GPU cards each node) with NVIDIA H200 GPUs (141GB memory each). The software stack is built on PyTorch 2.6, CUDA 12.4 (for H200), Transformer-Engine 2.4, and Triton 3.2.

We believe these efforts will enable researchers to reproduce our results and build upon our work.

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

# SUPPLEMENTARY MATERIALS

In the supplementary, we provide more details about our AIGCoder architecture and more implementation details. We organize our supplementary as follows.

- In Section A, we describe the usage of large language models in the writing process of this paper.
- In Section B, we provide the detailed formulation of our employed MoE architecture.
- In Section C, we provide more details about model specifications and parameter breakdown.
- In Section D, we provide more details of used dataset and implementation.
- In Section E, we provide more empirical analysis to show the effectiveness of our AIGCoder.
- In Section F, we provide more related work about memory-augmented LLMs.

## A LLM USAGE STATEMENT

We use LLM models solely as a tool to improve the clarity and fluency of the writing. All intellectual contributions, including research ideation and technical development, remain entirely our own. The LLMs did not contribute to the scientific content or conceptual direction of this work.

## B MORE DETAILS OF THE EMPLOYED MOE ARCHITECTURE

Below we provide the mathematical formulation of the Mixture-of-Experts (MoE) layer used in our model, which consists of two distinct components: a shared expert that processes all tokens and multiple routed experts that are selectively activated through a gating mechanism, which is similar with DeepseekMoE (Liu et al., 2024a). Given the input hidden states $\mathbf{U} \in \mathbb{R}^{L \times d}$, where $L$ represents the sequence length and $d$ denotes the hidden dimension, the MoE layer processes each token representation $\mathbf{u}_i$ (the $i$-th row of $\mathbf{U}$) independently through the following procedure.

**Formulation for Shared Expert.** For each input token representation $\mathbf{u} \in \mathbb{R}^d$, we first compute the logits for shared expert through:

$$\mathbf{h}^{\mathrm{S}} = \mathrm{MLP}^{\mathrm{S}}(\mathbf{u}), \tag{7}$$

where $\mathrm{MLP}(\cdot)$ denotes a multi-layer perceptron, $\mathbf{h}^{\mathrm{S}}$ remains the same shape, *i.e.*, $\mathbf{h}^{\mathrm{S}} \in \mathbb{R}^d$. Note that in our AIGCoder, we have only one shared expert. Then a sigmoid-gating mechanism is applied to modulate the shared experts' output, where the gating values are computed as:

$$\mathbf{g}^{\mathrm{S}} = \sigma(\mathbf{W}^{\mathrm{S}}\mathbf{u}), \tag{8}$$

where $\sigma(\cdot)$ denotes the sigmoid function, $\mathbf{g}^{\mathrm{S}}$ has the same shape with $\mathbf{h}^{\mathrm{S}}$, *i.e.*, $\mathbf{g}^{\mathrm{S}} \in \mathbb{R}^d$. The gated output of shared expert is then obtained by:

$$\mathbf{o}^{\mathrm{S}} = \mathbf{g}^{\mathrm{S}} \odot \mathbf{h}^{\mathrm{S}}. \tag{9}$$

**Formulation for Routed Experts.** For each input token representation $\mathbf{u} \in \mathbb{R}^d$, we first compute the router logits through:

$$\mathbf{g}^{\mathrm{R}} = \mathbf{W}^{\mathrm{R}}\mathbf{u}, \tag{10}$$

where $\mathbf{W}^{\mathrm{R}} \in \mathbb{R}^{d \times N_{\mathrm{route}}}$ denotes the router weight matrix, and $\mathbf{g}^{\mathrm{R}} \in \mathbb{R}^{N_{\mathrm{route}}}$ represents the logits for all routed experts. These logits are then passed through a sigmoid function to obtain gating values:

$$\mathbf{g}^{\mathrm{R}}_{\sigma} = \sigma(\mathbf{g}^{\mathrm{R}}), \tag{11}$$

where $\mathbf{g}^{\mathrm{R}}_{\sigma} \in \mathbb{R}^{N_{\mathrm{route}}}$ contains values in the range (0,1). These gating values are used to select the top-k experts for each token:

$$w_i = \frac{g^{\mathrm{R}}_{\sigma,i}}{\sum_{j \in \mathcal{T}} g^{\mathrm{R}}_{\sigma,j}}, \tag{12}$$

where $\mathcal{T}$ denotes the set of top-$k$ selected experts, and $w_i$ represents the normalized routing weight for the $i$-th expert. The output from the routed experts is computed as:

$$\mathbf{o}^{\mathrm{R}} = \left( \sum_{i \in \mathcal{T}} w_i \cdot \mathrm{Expert}_i(\mathbf{u}) \right), \tag{13}$$

where $\mathrm{Expert}_i(\cdot)$ denotes the $i$-th expert network. The final output $\mathbf{o}^{\mathrm{R}} \in \mathbb{R}^d$ has the same dimensionality as the input.

**Integration of Expert Outputs.** The final output for each token is obtained by combining the contributions from both the shared expert and the routed experts. The combined output is computed through element-wise addition:

$$\mathbf{o} = \mathbf{o}^{\mathrm{R}} + \mathbf{o}^{\mathrm{S}}, \tag{14}$$

where $\mathbf{o} \in \mathbb{R}^d$ represents the final output representation for the input token $\mathbf{u}$. This allows the model to benefit from both the consistently applied shared expertise and the specialized processing of the selectively activated routed experts. The complete output hidden states $\mathbf{O} \in \mathbb{R}^{L \times d}$ are constructed by applying this transformation to each token in the sequence independently.

## C    MORE DETAILS ON MODEL ARCHITECTURE

**Model Specifications**. In Table 4, we summarize the complete set of hyperparameters that define our model's architecture. We specify key design choices, including the total number of transformer layers, the dimensionality of the hidden states, the number of attention heads in each multi-head attention module, the resultant total parameter count, and the context length used for training.

Table 4: Configuration of proposed AIGCoder models across different scales. $d$, $d_c$ and $k$ denote hidden dimension, MLA latent dimension and LFA kernel size, respectively.

| Model | #Layers | $d$ | $d_c$ | $k$ | KMM ($F \times h \times d_u$) | Context Length | MoE Configuration |
|---|---|---|---|---|---|---|---|
| 1B | 16 | 1024 | 512 | 4 | 64×16×64 | 2048 | 1 shared + 8 experts |
| 5B | 20 | 2048 | 512 | 4 | 64×32×128 | 2048 | 1 shared + 8 experts |
| 7B | 28 | 2048 | 512 | 4 | 64×16×128 | 2048 | 1 shared + 8 experts |
| 13B | 28 | 2048 | 512 | 4 | 64×16×128 | 2048 | 1 shared + 16 experts |
| 33B | 18 | 4096 | 512 | 4 | 64×32×128 | 4096 | 1 shared + 16 experts |
| 60B | 32 | 4096 | 512 | 4 | 64×32×128 | 4096 | 1 shared + 16 experts |

**Component-wise Parameter Breakdown**. In Table 5, we present the distribution of parameters across different AIGCoder components. The MoE module consistently dominates the parameter budget, especially at larger scales, highlighting its central role in capacity expansion. In contrast, our newly introduced LFA kernel and KMM modules contribute only a marginal fraction of the total parameters, demonstrating the efficiency of these designs in enhancing model expressiveness without incurring substantial overhead. For completeness, we note that the input embedding and LM head are tied in the 1B, 5B, and 7B models, while they are untied in the 13B, 33B, and 60B models, with both components contributing equally when reported separately.

## D    MORE EXPERIMENTAL PROTOCOLS

### D.1    MORE DETAILS ON DATASET

In this section, we depict the details of the datasets used for pre-training and evaluation.

**Matrix Data Pile** (Zhang et al., 2024) is a large-scale, 4.5 trillion-token bilingual pre-training corpus meticulously curated for the MAP-Neo model series. The English subset is derived from a re-processing of high-quality public datasets, including RedPajama-Data-V2 (Weber et al., 2024), Dolma (Soldaini et al., 2024), Cultrax (EN) (Nguyen et al., 2024), Amber (Refined-Web) (Liu et al., 2024c), and SlimPajama (Cerebras, 2024), each subjected to a rigorous multi-phase filtering pipeline

Table 5: Distribution of parameters among different AIGCoder components across different model scales. For the 1B, 5B, and 7B models, the input embedding and the final LM head are tied, hence the LM head column is marked with "–". For the 13B, 33B, and 60B models, the embedding and LM head are untied, and their reported proportions correspond to each individual component.

| Model | Embedding | LFA Kernel | KMM | MLA | MoE | LM Head |
|-------|-----------|------------|-------|-------|--------|---------|
| 1B | 13.49% | 0.36% | 2.28% | 5.15% | 78.72% | – |
| 5B | 6.03% | 0.41% | 2.09% | 3.49% | 87.98% | – |
| 7B | 4.38% | 0.41% | 2.12% | 3.55% | 89.53% | – |
| 13B | 2.39% | 0.23% | 1.15% | 1.93% | 91.91% | 2.39% |
| 33B | 1.89% | 0.11% | 1.05% | 1.41% | 93.65% | 1.89% |
| 60B | 1.02% | 0.12% | 1.07% | 1.43% | 95.35% | 1.02% |

involving heuristic-based noise removal and deduplication strategies. The Chinese subset is predominantly composed of 80.6% newly crawled web content collected from scratch to address the persistent scarcity of open high-quality Chinese data; This foundational collection is enriched through integration of existing public Chinese datasets such as ChineseWebText (Chen et al., 2023), Wanjuan (He et al., 2023), Yayi2 (Luo et al., 2023), Cultrax (ZH) (Nguyen et al., 2024), and Skypile (Wei et al., 2023). To further enrich the corpus, high-quality data from diverse domains are incorporated, including programming code, academic papers (e.g., arXiv), books, government reports, and specialized collections for mathematics and science. All constituent data, regardless of origin or language, are processed through a unified framework that applies extensive language-specific cleaning (including HTML artifact removal and OCR error correction tailored for Chinese text) and robust deduplication techniques—encompassing exact document, MinHash LSH, and a proposed similar line deduplication—to ensure consistency, quality, and representativeness across the entire corpus.

**MMLU** (Hendrycks et al., 2021) is a comprehensive benchmark designed to evaluate multitask knowledge and reasoning capabilities of language models across 57 diverse subjects, spanning STEM, social sciences, humanities, and professional domains such as law, medicine, and business. The dataset consists of multiple-choice questions sourced from open-access exam materials, textbooks, and introductory college courses, ensuring broad coverage of both foundational academic principles and specialized domain-specific knowledge. The Questions are carefully curated to require not only factual recall, but also higher-order reasoning, including logical inference and conceptual understanding. To assess zero-shot and few-shot generalization, the benchmark is typically evaluated with minimal in-context examples, making it a robust measure of broad world knowledge and cross-domain transfer ability. Its design emphasizes domain diversity and task heterogeneity, providing a carefully structured assessment of model strengths and limitations across disciplines.

**C-Eval** (Huang et al., 2023) is a comprehensive Chinese benchmark designed to evaluate the knowledge and reasoning capabilities of foundation models across a wide range of academic disciplines. The dataset consists of 13,948 multiple-choice questions spanning 52 subjects, categorized into four broad domains: STEM, humanities, social sciences, and professional fields such as law, medicine, and finance. A key feature of C-Eval is its multi-level difficulty design, with questions sourced from exams at middle school, high school, college, and professional qualification levels, enabling fine-grained assessment of model performance across different expertise tiers. This hierarchical structure allows for a rigorous evaluation of both foundational knowledge and advanced reasoning skills in a Chinese educational context. To ensure data quality and minimize contamination, questions are collected from mock exams and past university tests, processed through careful parsing and manual validation, and formatted consistently with standardized answer choices. The benchmark supports both zero-shot and few-shot evaluation protocols, with a public development set provided for prompting and a private test set scored via an online submission system to maintain integrity. C-Eval also includes a challenging subset, C-Eval Hard, composed of complex subjects like advanced mathematics and physics that demand sophisticated problem-solving abilities. By covering diverse topics and difficulty levels within a culturally relevant framework, C-Eval provides a truly robust and nuanced evaluation of Chinese language understanding and domain-specific knowledge in LLMs.

**CMMLU** (Li et al., 2024) is a comprehensive benchmark designed to evaluate multitask language understanding and knowledge acquisition of large language models in the Chinese linguistic and cultural context. The dataset comprises 11,528 multiple-choice questions that span 67 subjects across

four broad categories: STEM, humanities, social sciences, and other domains, including 16 China-specific subjects such as Chinese driving rules, traditional food culture, and civil service exams. This deliberate inclusion of region-specific knowledge ensures that the assessment captures culturally grounded reasoning, distinguishing it from general-purpose multilingual benchmarks. Questions are curated from academic sources and standardized tests, covering difficulty levels from elementary to professional expertise, enabling fine-grained evaluation of model capabilities across diverse knowledge domains. To support the few-shot evaluation, each subject includes a development set of five exemplars, with the remainder forming the test set. CMMLU is specifically structured to assess zero-shot and few-shot generalization, making it a robust measure of a model's ability to transfer knowledge and reason across specialized and culturally relevant topics. Its design emphasizes domain diversity, cultural specificity, and task heterogeneity, providing a rigorous and well-nuanced assessment of Chinese language understanding that complements existing global benchmarks.

**HellaSwag** (Zellers et al., 2019) is designed to assess models' ability to predict future actions within common real-life situations. It comprises 39,905 context-based completion tasks drawn from video descriptions and textual narratives, each presenting four candidate endings—one correct human-authored option and three adversarially created distractors generated by models to closely match the context in plausibility. The challenge of HellaSwag lies in its emphasis on nuanced, grounded commonsense reasoning and its robustness against superficial linguistic cues or heuristic exploitation.

**ARC-Challenge** (Clark et al., 2018) is the more difficult subset of the AI2 Reasoning Challenge (ARC), designed to assess models' ability in nuanced scientific reasoning. It contains 2,590 multiple-choice questions from grade-school science curricula, filtered to exclude those easily answered by retrieval baselines. The dataset prioritizes tasks demanding deeper cognitive processing, compelling models to go beyond mere fact lookup and instead synthesize prior knowledge and reason about causal or explanatory relationships in scientific phenomena. Its construction emphasizes the distinction between simple information recall and genuine understanding of scientific concepts.

**HumanEval** (Chen et al., 2021) is a widely adopted benchmark designed to evaluate the functional correctness of the generated code through unit test execution. It features 164 carefully hand-crafted programming problems in Python, each accompanied by a function signature, a descriptive docstring, and multiple comprehensive ground-truth test cases that verify the output for various inputs. The evaluation relies on pass@1 metrics, which typically measures the probability that at least one generated solution (out of a single attempt) passes all associated test cases without error. A solution is deemed correct only if it produces semantically accurate results for every test instance, thereby enforcing strict robustness against syntactically valid but logically flawed code, a common failure mode in code generation. By covering algorithmic logic, string manipulation, and mathematical operations, the benchmark assesses a model's ability to generate precise, executable code from natural language specifications, making it a standard for measuring practical coding proficiency in LLMs.

**GSM8K** (Cobbe et al., 2021) is a benchmark for assessing the multi-step mathematical reasoning capabilities of language models through elementary school-level word problems. It contains 7,473 training and 1,319 hand-written test questions, each requiring two to eight sequential reasoning steps. The problems are designed to evaluate not only a model's ability to perform accurate arithmetic computations but also its capacity to decompose complex scenarios into coherent intermediate steps, applying concepts such as proportions, basic algebra, and numerical logic. Characterized by high linguistic diversity and real-world situational contexts, the dataset demands precise comprehension of structured natural language narratives. To facilitate transparent evaluation of reasoning processes, solutions are expected to include explicit chain-of-thought derivations, enabling analysis of both correct logical pathways and systematic errors, rather than merely predicting final answers.

### D.2 Supervised Fine-Tuning (SFT) Dataset Construction

To ensure high-quality supervision for instruction tuning, we construct a knowledge-intensive SFT dataset derived from large-scale raw corpora. The goal is to distill structured, factual question–answer pairs resembling benchmark tasks such as MMLU (Hendrycks et al., 2021), while rigorously preventing any benchmark leakage. Starting from 6.5 billion raw text segments, we obtain a final corpus of 2.5 million clean and decontaminated samples (average context length is about 4k with padding and masking). The entire pipeline comprises five major stages, as summarized below.

**Stage 1: Rule-based Filtering** (6.5B → 50M samples). We first perform large-scale heuristic filtering to remove irrelevant or low-quality content. This step identifies knowledge-oriented and interrogative sentences by matching question patterns (*e.g.*, what, why, which), detecting multiple-choice indicators (A. / B. / (A)(B) *etc.*), and applying keyword-based selection over 57+ academic domains (physics, law, medicine, history, etc.). Creative, subjective, or conversational text is excluded. Approximately 50 million candidate Q&A fragments remain after this stage.

**Stage 2: Semantic Retrieval Filtering** (50M → 5M samples). To retain samples that are related to benchmark-style knowledge, we apply an embedding-based retrieval filter. i) We first construct a knowledge seed query set by extracting 10,000 exam-style questions from MMLU, CMMLU, CEval, and GAOKAO, then distilling only their underlying knowledge concepts (*e.g.*, formula of Newton's second law) instead of reusing question text. ii) Both seed queries and candidate samples are encoded using the *bge-large-zh-v1.5* model, and top-k samples with cosine similarity > 0.65 are retained. **This retrieval step ensures topical relevance without reusing any benchmark text**.

**Stage 3: LLM-based Structuring and Validation** (5M → 3M samples). In this stage, an LLM (*e.g.*, Qwen-Max) is prompted to validate and normalize each candidate fragment into a structured format containing {question, answer, subject, type}. The model verifies factual correctness, conciseness, and clarity, discarding vague or subjective responses. Optional distractor options are added for multiple-choice questions. The process yields approximately 3 million high-quality QA samples.

**Stage 4: Balancing and Standardization** (3M → 2.8M samples). We perform stratified sampling to balance the subject distribution according to the proportions in MMLU and CMMLU. Underrepresented disciplines such as ethics and nutrition are slightly oversampled. All samples are standardized into a consistent instruction–input–output format and de-duplicated using MinHash and LSH. Personally identifiable information (PII) is also filtered. The resulting dataset contains about 2.8 million balanced and clean samples.

**Stage 5: Benchmark Decontamination** (2.8M → 2.5M samples). We employ a rigorous three-fold process to eliminate any potential benchmark leakage.

- **Literal Matching**: remove exact or near-duplicate overlaps (edit distance ≤5 or Jaccard similarity ≥0.95) with a blacklist containing all questions and answers from MMLU, CMMLU, CEval, GAOKAO, and AGIEval (about 20–30k items).

- **Semantic Matching**: compute cosine similarity between encoded blacklist items and candidates; remove those with similarity ≥0.92.

- **Knowledge Triplet Matching**: extract structured knowledge triples (entity, relation, value) using an LLM and flag samples that share identical triples with blacklist items as high-risk candidates for benchmark leakage. These high-risk samples are then either manually audited or conservatively removed. As an auxiliary probe, we also perform generative leakage detection with a smaller trained model, which flags samples whose predicted answers match the ground-truth with extremely high confidence (e.g., probability > 0.99) as additional high-risk candidates. Finally, a small-scale manual audit of 1,000 random samples confirms a residual leakage rate below 0.1%. The resulting dataset contains 2.5 million high-quality SFT samples.

This multi-stage pipeline has been validated across several large-model projects and demonstrates strong practicality: it efficiently distills high-quality, knowledge-focused QA data while meeting the strict decontamination and transparency standards required for academic evaluation and open-source release. It provides a clean and reliable basis for the supervised fine-tuning phase of our AIGCoder.

### D.3 MORE IMPLEMENTATION DETAILS

**Hyper-parameters Setting**. All models in the AIGCoder series (1B, 5B, 7B, 13B, 33B, and 60B) are trained using the AdamW optimizer with momentum parameters $\beta_1 = 0.9$, $\beta_2 = 0.95$, and a weight decay of 0.1 to ensure stable and efficient convergence. The learning rate is carefully tuned for each model scale and training phase. For pre-training, we use an initial learning rate of $8 \times 10^{-3}$ for the 1B model, and $3 \times 10^{-4}$ for the 5B and 7B models, decreasing to $2 \times 10^{-4}$ for the 13B model, and further to $1 \times 10^{-4}$ for the 33B and 60B models. During supervised fine-tuning (SFT),

a lower learning rate is applied: $3 \times 10^{-4}$ for the 1B model, $4 \times 10^{-5}$ for 5B and 7B, $2 \times 10^{-5}$ for 13B, and $1 \times 10^{-5}$ for 33B and 60B. All models are trained using mixed-precision with gradient clipping set to 1.0. The context length during training ranges from 2048 for smaller models to 4096 for the larger variants, as detailed in Section C. All main pre-training experiments use a global batch size of 16,384 and run for a fixed duration of 10,000 steps, consuming hundreds of billions of tokens depending on model context length. For ablation studies, a smaller batch size of 1,024 is used.

**SFT Alignment Strategy.** We emphasize that AIGCoder models are aligned purely through supervised fine-tuning (SFT) on our curated large-scale instruction dataset described in Section D.2. No reinforcement learning (RL) or preference optimization algorithms (e.g., RLHF, GRPO) are employed at any stage. All alignment behaviors, including instruction following and factual consistency, emerge from supervised training on the cleaned and decontaminated SFT corpus, which contains diverse knowledge-intensive question–answer and multi-turn instruction samples across more than 50 domains. This design choice ensures a fully transparent and reproducible alignment process without reliance on external reward models.

**Details of Parallelism Strategy**. For models scales up to and including 13B parameters, we shard the parametric memory with expert parallelism (EP=2) and otherwise use pure data parallel (DP) replication; pipeline, tensor, and sequence parallelism are disabled. Training runs on 20 nodes with 8 Ascend 910B GPUs each, which meets memory and throughput targets without intra-layer model parallelism. For the 30B and 60B model, we employ an 8 stage pipeline (PP=8) with with expert parallelism of degree four (EP=4) and data parallelism (DP) over the residual dimension; tensor/sequence parallelism are unused. Using Megatron-LM with Transformer Engine, we train stably on H200, validated on a 4-node setup and scaled to a 50-node cluster ($8\times$ H200 per node).

### D.4 MORE DETAILS OF VISUALIZATION OF KNOWLEDGE FIELD-DOMAIN ASSOCIATIONS

To further elucidate the internal mechanisms of the proposed architecture, we conducted a comprehensive visualization analysis (please refer to Figures 2 and 6) to verify whether the Knowledge Block spontaneously acquires structured, domain-specific representations. The primary objective was to observe if high-relevance tokens from distinct domains focus on different key regions via the cross-attention mechanism. Architecturally, a trainable global storage module containing initialized Key-Value pairs ($K_{kb}, V_{kb} \in \mathbb{R}^{N \times d}$, where $N$ denotes the number of knowledge slots) is integrated into the MoE-MLA framework. In this setup, a query vector $Q_{ctx}$ is derived from the intermediate compressed KV representations via a lightweight MLP, facilitating a cross-attention operation defined as:

$$\text{Attention}(Q_{ctx}, K_{kb}, V_{kb}) = \text{softmax}\left(\frac{Q_{ctx}K_{kb}^{\top}}{\sqrt{d}}\right)V_{kb} \qquad (15)$$

The resulting output subsequently serves as supplementary input for the MoE experts and the router. For the experimental setup, we utilized multiple sub-domain test sets from the MMLU benchmark, spanning disciplines such as History, Mathematics, Physics, Chemistry, and Politics. For each input sample, we identified high-relevance tokens—such as "Newton," "Calculus," or "Constitution"— using domain dictionary matching and embedding similarity annotation. We then extracted the attention score vectors generated between $Q_{ctx}$ and $K_{kb}$ for these tokens, normalizing them to obtain a probability distribution $\alpha_i \in \mathbb{R}^N$, which represents the focus intensity on specific knowledge slots. The core observations reveal a distinct pattern: high-relevance tokens from different domains exhibit attention scores that are significantly concentrated on disparate subsets of keys. For instance, mathematical tokens predominantly activate specific key indices (e.g., #12, #45, #89), whereas historical tokens focus on a separate set of indices (e.g., #3, #67, #102). Although related fields like Physics and Chemistry show partial overlap, they maintain exclusive high-activation regions. These findings empirically validate that the Knowledge Block possesses the capability to automatically organize domain knowledge into structured partitions without explicit supervision, thereby enabling the model to perform precise, domain-aware retrieval via the query mechanism to distinct knowledge regions.

## E MORE DISCUSSIONS

**Comparisons with Sliding-window Attention**. While both LFA and sliding-window attention introduce locality, they differ fundamentally in design and purpose: i) **No truncation of global con-**

**text**. Sliding-window attention limits each token to a fixed local window by masking, which prevents modeling of long-range dependencies. Instead, our LFA preserves full attention and adds a lightweight convolutional fusion before attention, enriching local representations without restricting the receptive field. ii) **Learnable locality rather than fixed windows**. In sliding-window attention, the local region is predefined and static. In LFA, the convolutional kernels are learned jointly with the model, allowing flexible adaptation of how local context is fused across layers and heads. iii) **Complementary rather than substitutive**. Sliding-window attention replaces part of attention computation to save FLOPs, while LFA complements attention by providing locally enhanced inputs. It can thus be seamlessly combined with other attention variants without altering attention sparsity or complexity.

**Throughput and Training Efficiency Analysis**. While the convergence speed in the main text is reported in terms of training steps, a complete efficiency evaluation must also consider the computational cost per step. To this end, we compare the training throughput (in tokens per second) of AIGCoder against the baseline without LFA and KMM under identical hardware configurations (8×NVIDIA A100 GPUs, 80GB memory). The measured throughput for the baseline is 975 tokens/second, while AIGCoder achieves a throughput of 950 tokens/second. This analysis confirms that the per-step computational overhead introduced by our novel modules is minimal (approximately 2.6%). Therefore, the significant reduction in the number of training steps required to reach the target loss, as reported in the main text, translates directly into a substantial reduction in total wall-clock training time. This underscores the training efficiency of AIGCoder architecture.

**More Visualization of Knowledge Field-Domain Associations**. In Figure 6, we present the field-domain associations for the proposed KMM with 64 knowledge fields across layers 8, 12, 16, 20, 24 and 28. These visualizations, generated using the same experimental configuration as the main manuscript, show that individual fields spontaneously evolve to represent domain-specific concepts through end-to-end training. These findings are consistent with the observations in the main text.

**Knowledge Field Editability Analysis**. To further substantiate the claim that the Knowledge Memory Module (KMM) stores explicit, interpretable, and editable domain-level knowledge, we conduct a set of knowledge-field removal experiments. For each selected knowledge field in Layer 4 (the layer visualized in Figure 2), we zero out its value parameters and compare the model's performance on five representative MMLU domains with and without this field.

If a field truly encodes domain-specific knowledge, removing it should primarily hurt the corresponding domain while leaving other domains almost unchanged. This is exactly what we observe in Tables 6-10: removing field 32 mainly degrades algebra, field 18 chemistry, field 51 physics, and fields 24/25 politics and history, with only small fluctuations on unrelated domains. These domain-selective degradations provide strong empirical evidence that KMM organizes knowledge into localized, disentangled memory slots that can be directly manipulated.

**Analysis of Cross-Field Interference via Cosine Similarity** To directly examine whether different knowledge fields interfere with each other or collapse to similar representations, we analyze the geometry of the learned KMM keys. For the layers visualized in Figures 2 and 6, we take all $F = 64$ key vectors of the knowledge fields and compute the pairwise cosine similarity matrix between field pairs. If multiple fields collapsed or encoded heavily overlapping content, we would expect large off-diagonal cosine values and visible clustered structures in this similarity heatmap.

The resulting cosine-similarity matrix (see Figures 7-13) is very close to an identity matrix. For instance, in layer 4, the mean off-diagonal cosine similarity is -0.001, and the maximum absolute off-diagonal cosine similarity is only 0.0513. In other words, different knowledge fields are nearly orthogonal in the learned key space, with no evident clustering or collapse. This indicates that, at $F = 64$, the KMM learns a set of well-separated, specialized fields rather than redundant or interfering ones. While this does not preclude interesting behaviors at much larger scales, it provides concrete empirical evidence that cross-field interference and field collapse are not observed in our current setting, supporting the expressiveness and stability of the proposed memory module.

## F  MORE RELATED WORK

**Memory-Augmented LLMs**. To overcome the limitations of fixed context windows, memory-augmented approaches integrate external or historical knowledge to enable long-context model-

Table 6: Performance comparison with and without knowledge field 32 in Layer 4. According to Figure 2 in the main paper, field 32 is strongly associated with *algebra* knowledge. We set the parameters of this field in Layer 4 to the zero vector and re-evaluate five MMLU domains.

| Domain | Original | Ablated | Relative Drop |
|--------|----------|---------|---------------|
| algebra | 85 | 58 | -31.8% |
| chemistry | 86 | 75 | -12.8% |
| physics | 92 | 81 | -12.0% |
| politics | 93 | 92 | -1.1% |
| history | 95 | 96 | +1.1% |

Table 7: Performance comparison with and without knowledge field 18 in Layer 4. According to Figure 2 in the main paper, field 18 is strongly associated with *chemistry* knowledge. We set the parameters of this field in Layer 4 to the zero vector and re-evaluate five MMLU domains.

| Domain | Original | Ablated | Relative Drop |
|--------|----------|---------|---------------|
| algebra | 85 | 81 | -4.7% |
| chemistry | 86 | 62 | -27.9% |
| physics | 92 | 87 | -5.4% |
| politics | 93 | 90 | -3.2% |
| history | 95 | 91 | -4.2% |

ing and dynamic updating. Explicit memory stores knowledge in human-readable forms—such as text, summaries (Zhong et al., 2024), or structured knowledge graphs (Jimenez Gutierrez et al., 2024)—and relies on symbolic retrieval. ChatDB (Hu et al., 2023) uses SQL queries over relational databases, while MemGPT (Packer et al., 2023) emulates virtual memory with LLM-generated function calls to page data between context and storage. MemLLM (Modarressi et al., 2025) fine-tunes models to query a knowledge graph-based triple memory. Mem0 (Chhikara et al., 2025) introduces a production-oriented memory system that extracts user facts into graph-structured memories from conversations for multi-hop reasoning. In contrast, implicit memory encodes information compactly in latent vectors. The Memorizing Transformer (Wu et al., 2022) and LONGMEM (Wang et al., 2023) cache key-value pairs and retrieve them via approximate kNN search to extend attention. MemoryLLM (Wang et al., 2024) maintains per-layer memory pools updated through self-editing with random dropping for controlled growth. M+ (Wang et al., 2025b) further separates short-term (GPU) and long-term (CPU) memory, using a co-trained retriever to access over 160K tokens with minimal GPU overhead.

Our Knowledge Memory Module (KMM), as detailed in Section 4.3, is not designed for context extension or external memory augmentation—the two primary goals of existing memory-augmented LLMs. Instead, KMM is a novel architectural component that explicitly decouples global knowledge storage from local computation, thereby addressing the implicit coupling of knowledge and computation inherent in standard LLMs. Specifically, KMM introduces a lightweight, parameterized memory component that stores domain-level knowledge in learnable key–value fields. These fields are trained end-to-end but remain fixed during inference, enabling direct, differentiable access to structured global knowledge without online updates.

Table 8: Performance comparison with and without knowledge field 51 in Layer 4. According to Figure 2 in the main paper, field 51 is strongly associated with *physics* knowledge. We set the parameters of this field in Layer 4 to the zero vector and re-evaluate five MMLU domains.

| Domain | Original | Ablated | Relative Drop |
|---|---|---|---|
| algebra | 85 | 82 | -3.5% |
| chemistry | 86 | 81 | -5.8% |
| physics | 92 | 62 | -32.6% |
| politics | 93 | 92 | -1.1% |
| history | 95 | 93 | -2.1% |

Table 9: Performance comparison with and without knowledge field 24 in Layer 4. According to Figure 2 in the main paper, field 24 is strongly associated with *politics* knowledge. We set the parameters of this field in Layer 4 to the zero vector and re-evaluate five MMLU domains.

| Domain | Original | Ablated | Relative Drop |
|---|---|---|---|
| algebra | 85 | 85 | 0.0% |
| chemistry | 86 | 84 | -2.3% |
| physics | 92 | 90 | -2.2% |
| politics | 93 | 75 | -19.4% |
| history | 95 | 83 | -12.6% |

Table 10: Performance comparison with and without knowledge field 25 in Layer 4. According to Figure 2 in the main paper, field 25 is strongly associated with *history* and *politics* knowledge. We set the parameters of this field in Layer 4 to the zero vector and re-evaluate five MMLU domains.

| Domain | Original | Ablated | Relative Drop |
|---|---|---|---|
| algebra | 85 | 78 | -8.2% |
| chemistry | 86 | 80 | -7.0% |
| physics | 92 | 89 | -3.3% |
| politics | 93 | 71 | -23.7% |
| history | 95 | 69 | -27.4% |

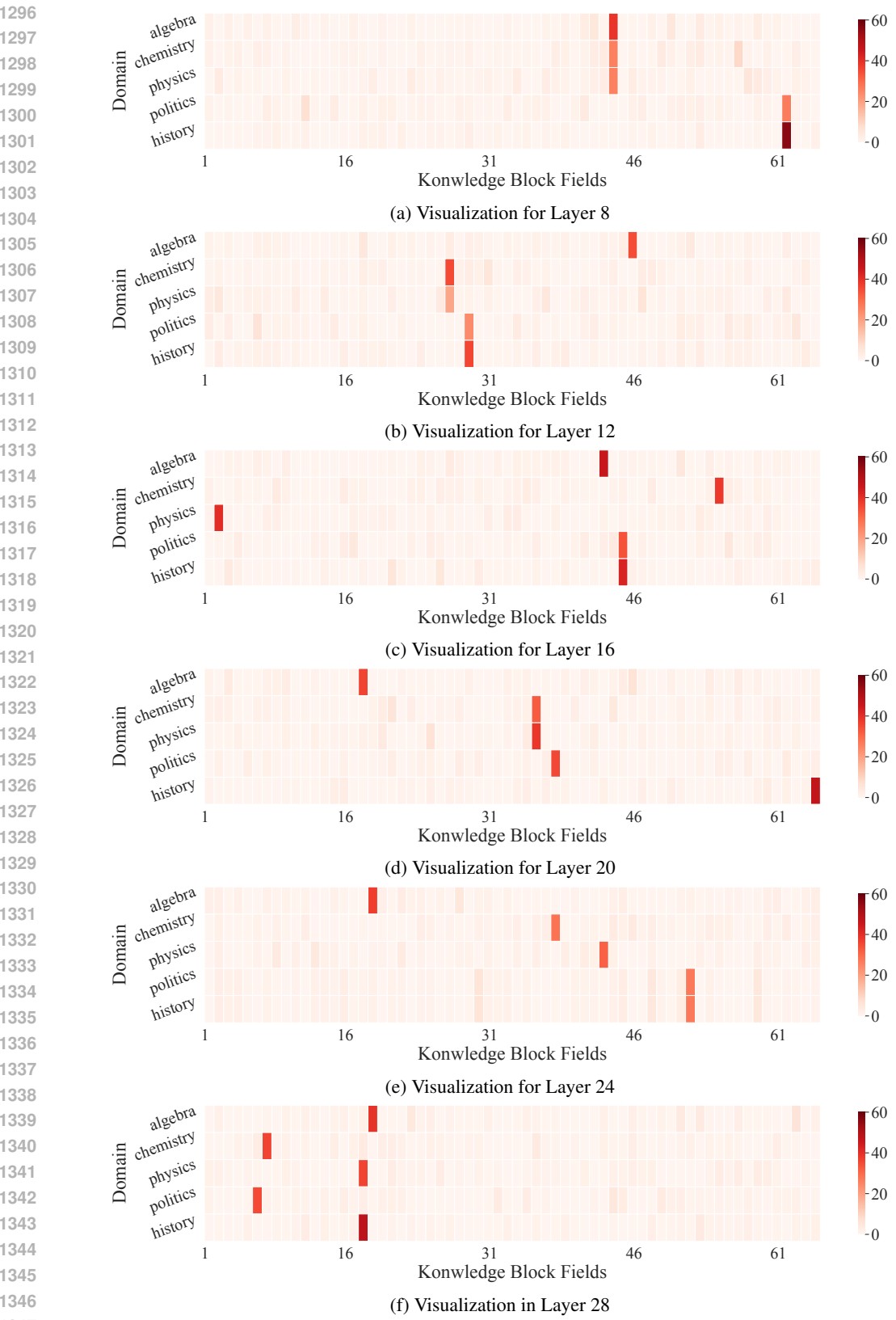

(a) Visualization for Layer 8

(b) Visualization for Layer 12

(c) Visualization for Layer 16

(d) Visualization for Layer 20

(e) Visualization for Layer 24

(f) Visualization in Layer 28

Figure 6: Field–domain associations of the proposed KMM with 64 knowledge fields, evaluated on five MMLU domains (1,024 samples each). The heatmap of $\text{softmax}(\mathbf{H}\mathcal{K}/d_u)$ shows that fields emerge with domain-specific specialization, enabling explicit and interpretable knowledge retrieval.

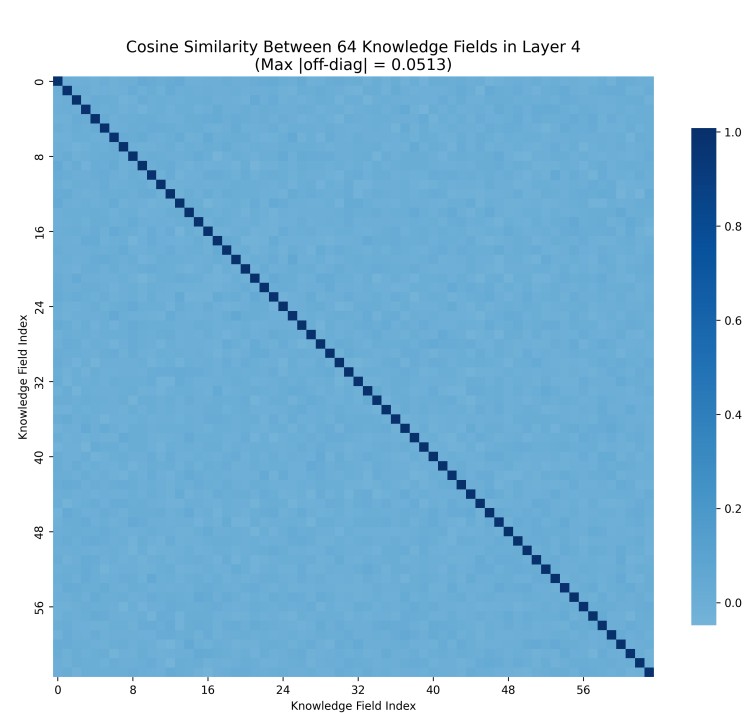

Figure 7: Cosine similarity matrix between 64 KMM knowledge fields in Layer 4.

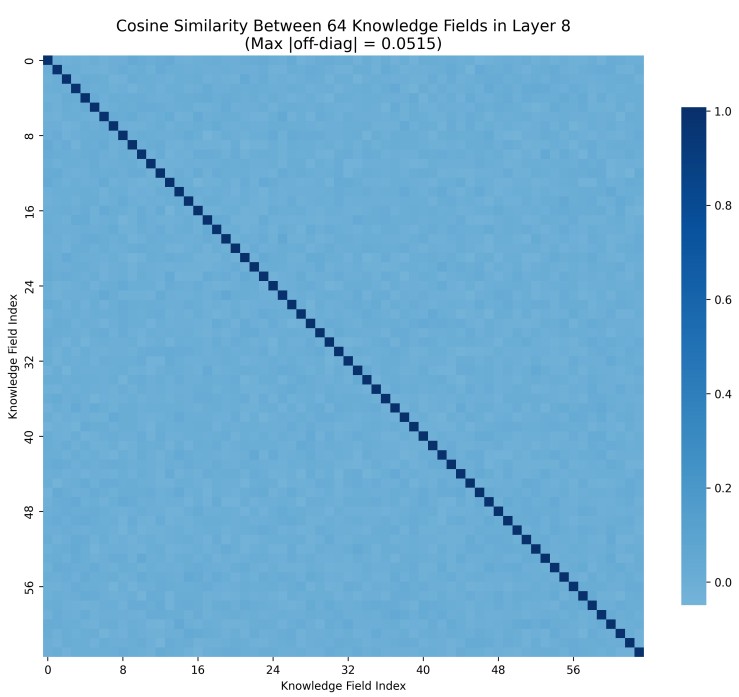

Figure 8: Cosine similarity matrix between 64 KMM knowledge fields in Layer 8.

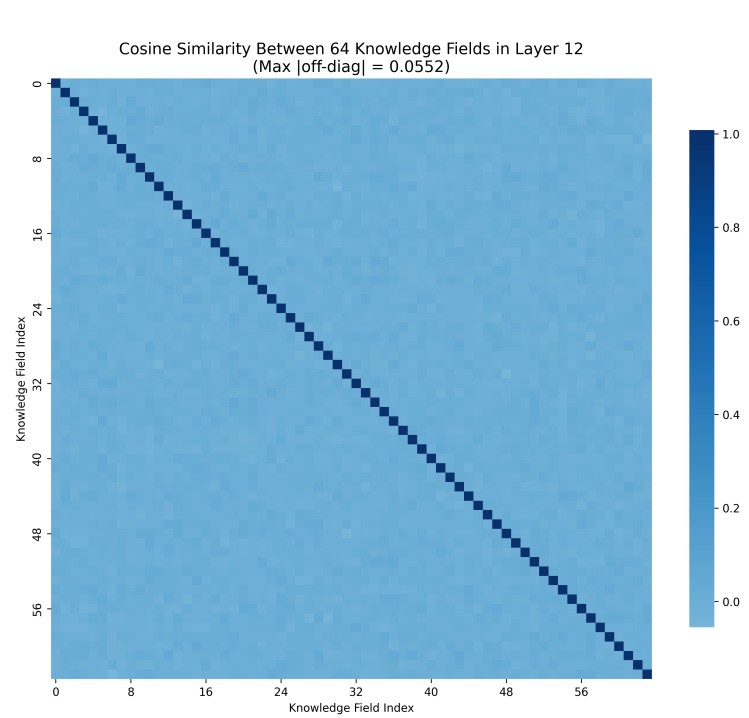

Figure 9: Cosine similarity matrix between 64 KMM knowledge fields in Layer 12.

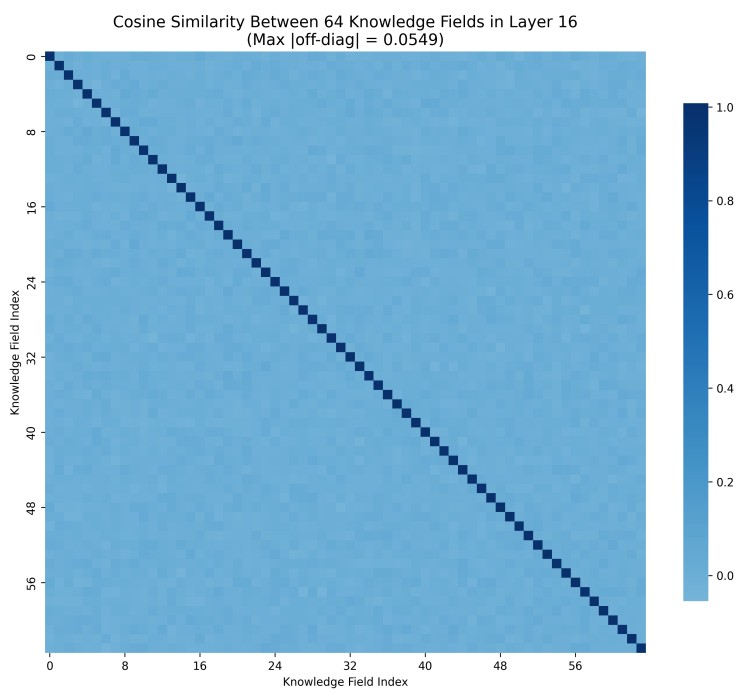

Figure 10: Cosine similarity matrix between 64 KMM knowledge fields in Layer 16.

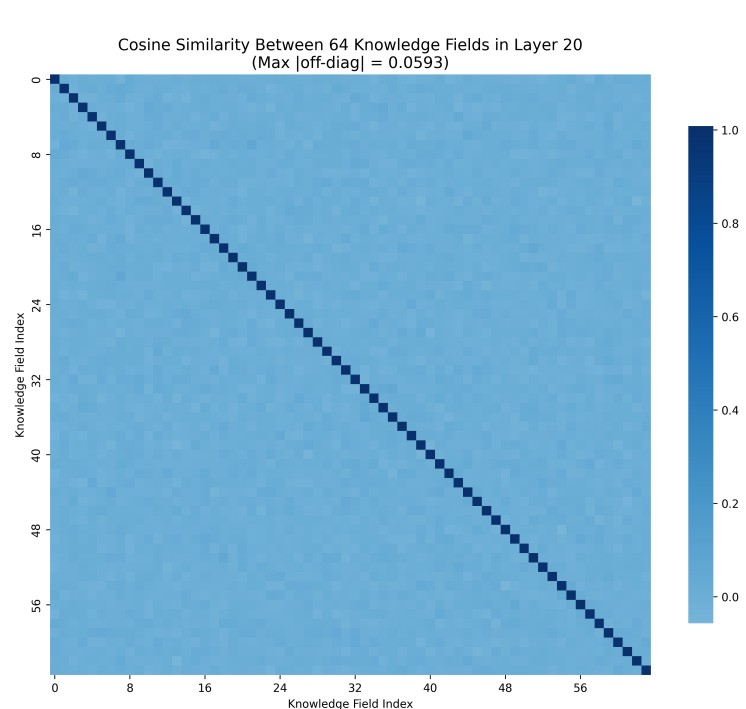

Figure 11: Cosine similarity matrix between 64 KMM knowledge fields in Layer 20.

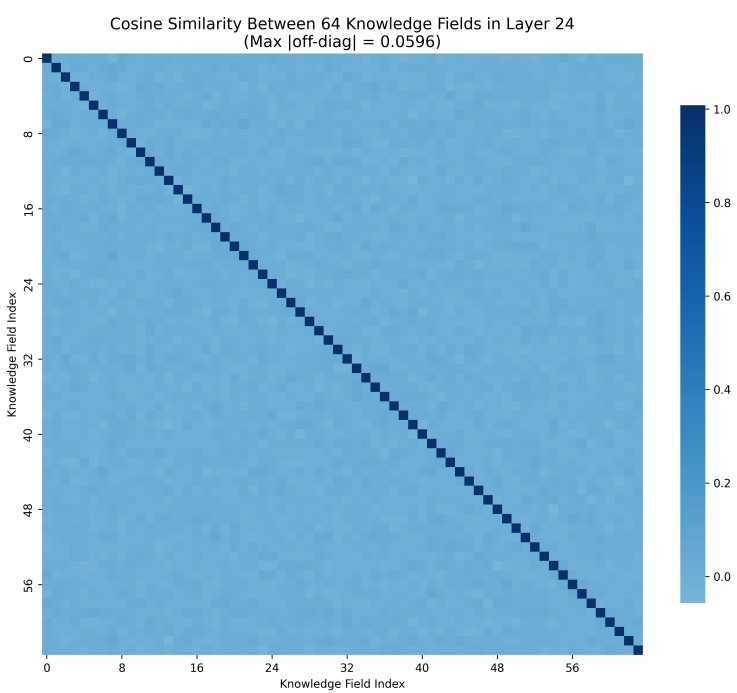

Figure 12: Cosine similarity matrix between 64 KMM knowledge fields in Layer 24.

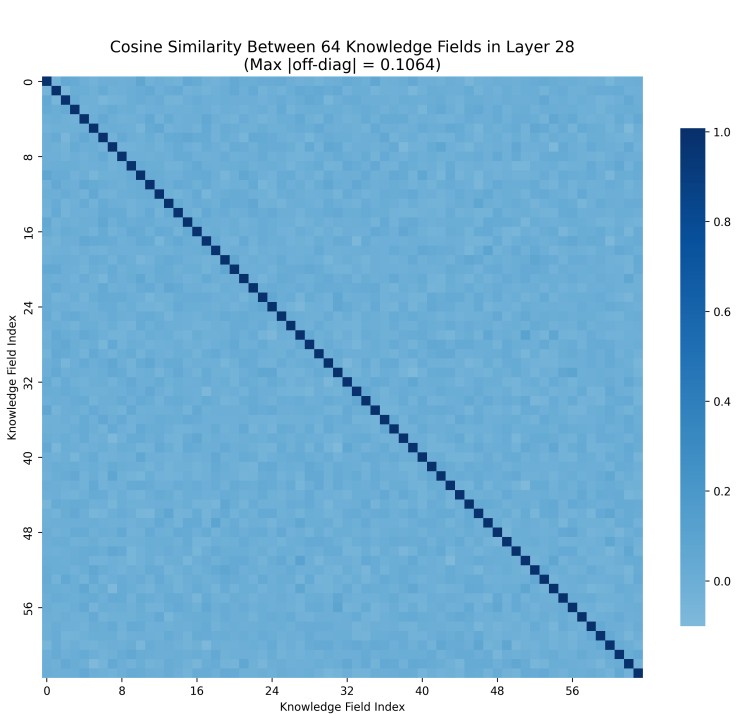

Figure 13: Cosine similarity matrix between 64 KMM knowledge fields in Layer 28.

