# OpenReview forum: "AIGCoder 1.0: Locally-Enhanced Language Modeling with Explicit and Structured Knowledge Memory"
_ICLR.cc/2026/Conference — Submitted to ICLR 2026_

### Official Review · Reviewer_MoAF · 2025-10-31

**Soundness:** 3
**Presentation:** 3
**Contribution:** 2
**Rating:** 2
**Confidence:** 4

**Summary:**

The paper addresses two fundamental issues with current LLM architectures: (1) the lack of explicit inductive bias in the attention mechanism to address local structures in the input sequence. (2) the lack of decoupling between knowledge extraction and forward computation path, or in other words, reliance on knowledge stored in the model parameters. The paper suggests mechanisms to address those issues and to integrate the two levels of knowledge (local, sequence-dependent and external, sequence-independent) in an efficient way.  They do that by adding two new modules to existing architectures: Local Fusion attention (LFA) and Knowledge Memory Module (KMM). LFA uses a convolution operation with a learnable kernel to aggregate local information in each token representation. KMM stores keys and values in an external memory which are retrieved through querying during inference. The final representation which goes into the attention layer is a combination (addition) of these two representations. Experiments show the advantage of their approach compared to baseline models in terms of overall performance and number of training steps.

**Strengths:**

The paper addresses fundamental issues in LLM architectures. Overall, it is well-written and fairly easy to read except for some details missing in the description of KMM. The authors conducted extensive experimentation against baseline models as well as ablations.

**Weaknesses:**

Incorporating explicit inductive biases into models is a tricky business. On the one hand, it can improve model training and efficiency, on the other, it may be limiting the learning process and generalization. One should have a very clear intuition backed by evidence for including inductive bias. In the case of locality, the authors argue that current architectures are not incorporating long and short range dependencies in the most efficient way because all tokens are treated equally. However, there is evidence that LLMs do obtain implicit recency (locality) bias through pre-training. Furthermore, for different data and different tasks (e.g. natural language vs coding), we would expect a different balance between long and short range dependencies. This begs the question: why should we add an explicit locality bias rather than let the model figure it on its own?

I am surprised that there is no mention at all of previous works on memory-augmented models and their relation to KMM. I'd expect to see such a discussion in the related work section as well as comparative experiments.

**Questions:**

- Lines 52-53 "This may lead to suboptimal…” I don’t understand this sentence.

- In what sense are current architectures inefficient in integrating local and external knowledge (lines 49-50)? Do you have evidence for that?

- Why is LFA better than sliding window attention?

- What do we need MLA for? If I understand correctly, you can apply your techniques directly on vanilla attention. I understand that it makes things more efficient, but for the presentation of your ideas, why add another unnecessary level of complexity?

- I’m missing details on how K and V in Equation 5 are being updated during training.

- 174-177: repeated sentence

- Please provide more details on how the visualization experiment (Figure 2) was done.

- The visualization experiment shows that some memory fields are strongly associated with specific domains. However, this happens in a relatively small fraction of the fields. Could this be just by chance? Have you conducted an analysis to rule that out?

- Lines 430-431: "These show that increasing the kernel size consistently lowers training”: I wouldn’t make such a statement based on only two kernel sizes. Clearly as k gets closer to the entire sequence length, we don’t expect to see a benefit compared to regular attention.

---

> ### Author Response · Authors · 2025-11-22
> **Responses to Reviewer MoAF [1/4]**
>
> We are grateful for your time and effort. We would like to answer your questions below.
>
> ---
>
> >Q1. Incorporating explicit inductive biases into models is a tricky business. On the one hand, it can improve model training and efficiency, on the other, it may be limiting the learning process and generalization. One should have a very clear intuition backed by evidence for including inductive bias. In the case of locality, the authors argue that current architectures are not incorporating long and short range dependencies in the most efficient way because all tokens are treated equally. However, there is evidence that LLMs do obtain implicit recency (locality) bias through pre-training.
>
> **A1.** We agree that inductive biases must be added carefully. Our goal with LFA is not to *restrict* what the model can learn, but to provide a **learnable local prior** that improves efficiency while preserving full expressivity.
>
> **Why an explicit local bias does not limit generalization**. LFA does not **mask/sparsify attention** and does not remove any long-range paths; the full attention mechanism remains intact. LFA adds a lightweight group convolution before attention to **densify short-range features** so that attention receives richer local representations and can allocate more capacity to long-range structure, rather than redundantly relearning local patterns token-by-token. We will add dedicated long-range evaluations in the next few days to explicitly verify this effect.
>
>
> **Empirical Evidence that LFA helps**. In the ablations (Figure 4 of the manuscript), adding LFA module (with the attention otherwise unchanged) yields **faster convergence (1.11×)** and **lower validation perplexity** than the baseline under identical training settings (e.g., 31.82 → 30.88 PPL), indicating better modeling with no sign of over-constraint.
>
> ---
>
> >Q2. Furthermore, for different data and different tasks (e.g. natural language vs coding), we would expect a different balance between long and short range dependencies. This begs the question: why should we add an explicit locality bias rather than let the model figure it on its own?
>
> **A2.** We appreciate the reviewer's insightful question. Indeed, the balance between short- and long-range dependencies varies across domains, but our design of **Local Fusion Attention (LFA)** is intentionally *adaptive* rather than fixed.
>
> **Why not let the model learn it on its own**. While large models can internalize a recency bias during pre-training, relying solely on full pairwise attention is computationally inefficient. at every layer/head, attention must repeatedly reconstruct short-range patterns from scratch. LFA supplies a lightweight, learnable convolutional cue that pre-aggregates local context into latent features before attention, so each token enters attention with richer, locally fused features. This raises the per-layer representational quality along the local dimension, letting attention focus more on pattern selection and non-local composition rather than re-discovering n-gram/proximal structure each time—without masking or sparsifying attention, so all long-distance paths are preserved.
>
> **Adaptivity across data types.** LFA adapts to different tasks **via its learned group-wise convolution parameters** (per head/group), which determine *how* local evidence is fused before attention—even though the kernel size k is fixed. The subsequent attention remains full-range, so long-distance dependencies are preserved. Ablations show that a moderate and aligning conv group size with token dimension yield consistent PPL gains, indicating effective, task-adaptive local fusion without overfitting (Figures 5a–b of the manuscript).

---

> > ### Author Response · Authors · 2025-11-25
> > **Looking forward to the response from Reviewer MoAF**
> >
> > Dear Reviewer MoAF,
> >
> > We have tried our best to address all the concerns and provided explanations to all questions. We sincerely hope that our answer has addressed your initial concerns. Kindly let us know if you have any other concerns, and we will do our best to address them.
> >
> > Best regards,
> >
> > The Authors

---

> ### Author Response · Authors · 2025-11-22
> **Responses to Reviewer MoAF [2/4]**
>
> >Q3. I am surprised that there is no mention at all of previous works on memory-augmented models and their relation to KMM. I'd expect to see such a discussion in the related work section as well as comparative experiments.
>
>
> **A3.** We clarify that while our Knowledge Memory Module (KMM) shares the high-level goal of enhancing knowledge access with prior memory-augmented models, its purpose, design, and integration are fundamentally different. Existing approaches such as Memorizing Transformer\[r1\], LONGMEM\[r2\], MemoryLLM\[r3\], and M+\[r4\] primarily aim at context extension by caching past activations or external content to overcome fixed-length context limitations. Similarly, explicit memory systems like MemGPT\[r5\], ChatDB\[r6\], and Mem0\[r7\] treat memory as an external symbolic store, accessed via function calls or queries, often for user-specific state or long-term personalization.
>
> In contrast, KMM is not designed for context extension or external memory augmentation—the two primary goals of existing memory-augmented LLMs. Instead, KMM is a **novel architectural component that explicitly decouples global knowledge storage from local computation**, thereby addressing the implicit coupling of knowledge and computation inherent in standard LLMs. Specifically, KMM introduces a lightweight, parameterized memory component that stores domain-level knowledge in learnable key–value fields. These fields are trained end-to-end but **remain fixed during inference**, enabling direct, differentiable access to structured global knowledge without online updates.
>
> We have discussed and cited above memory-argumented LLMs works in Section F of the supplementary.
>
> \[r1\] Memorizing Transformers. ICLR 2022.
>
> \[r2\] Augmenting language models with long-term memory. NeurIPS 2023.
>
> \[r3\] MEMORYLLM: Towards Self-Updatable Large Language Models. ICML 2024.
>
> \[r4\] M+: Extending MemoryLLM with Scalable Long-Term Memory. ICML 2025.
>
> \[r5\] MemGPT: Towards LLMs as Operating Systems. arXiv 2023.
>
> \[r6\] ChatDB: Augmenting LLMs with Databases as Their Symbolic Memory. arXiv 2023.
>
> \[r7\] Mem0: Building production-ready ai agents with scalable long-term memory. arXiv 2025.
>
> ---
>
> >Q4. Lines 52-53 "This may lead to suboptimal…” I don’t understand this sentence.
>
>
> **A4.** We thank the reviewer for pointing this out. The sentence in Lines 52–53 was meant to highlight an **efficiency** issue rather than a functional limitation of attention. Self-attention can indeed capture local dependencies, but it does so by computing all pairwise token interactions, including many redundant ones between adjacent tokens that could be modeled more directly. This redundancy slows learning of short-range patterns and increases computational cost. Our Local Fusion Attention addresses this by adding a lightweight convolutional fusion before attention, providing the same representational capacity but improved efficiency.
>
> We will revise the sentence to make this clearer in Section 1:
>
> > ''For modeling sequence-internal local information, while attention mechanisms can capture both short- and long-range dependencies, it does so through redundant full pairwise interactions, which makes learning local patterns computationally inefficient.''
>
> ---
>
> Q5. In what sense are current architectures inefficient in integrating local and external knowledge (lines 49-50)? Do you have evidence for that?
>
> **A5.** Our statement that ''current architectures are inefficient in integrating local and external knowledge'' refers to **architectural inefficiency**, not to empirical failure.
>
> Current LLMs process all dependencies uniformly through full self-attention and store global knowledge implicitly within expert parameters (in MoE) or dense weights. This coupling makes short-range pattern modeling redundant and knowledge access indirect. Prior work has also pointed out these limitations—for example, xxx.
>
> **Empirical evidence**. Our empirical results further support this interpretation: adding LFA and KMM—modules that explicitly separate local fusion and global knowledge retrieval—achieves **the same validation loss in 7.5k vs. 10k steps (1.33× faster)** under identical training settings. Crucially, this speedup is obtained at **nearly the same parameter count** (LFA adds only **~0.41%** params)**, so total wall-clock time and compute are substantially reduced. We also observe **lower pre-training perplexity** and **interpretable specialization** of knowledge fields (Figure 2 of the manuscript), indicating that explicit mechanisms for local and global information improve **computational efficiency**.

---

> ### Author Response · Authors · 2025-11-22
> **Responses to Reviewer MoAF [3/4]**
>
> >Q6. Why is LFA better than sliding window attention?
>
> **A6.** We thank the reviewer for the question. While both LFA and sliding-window attention introduce locality, they differ fundamentally in design and purpose:
>
>  - **No truncation of global context.** Sliding-window attention limits each token to a fixed local window by masking, which prevents modeling of long-range dependencies. Instead, our LFA preserves *full attention* and adds a lightweight convolutional fusion *before* attention, enriching local representations without restricting the receptive field.
>  - **Learnable locality rather than fixed windows.** In sliding-window attention, the local region is predefined and static. In LFA, the convolutional kernels are *learned* jointly with the model, allowing flexible adaptation of how local context is fused across layers and heads.
>  - **Complementary rather than substitutive.** Sliding-window attention replaces part of attention computation to save FLOPs, while LFA complements attention by providing locally enhanced inputs. It can thus be seamlessly combined with other attention variants without altering attention sparsity or complexity.
>
> We have included thse discussions in Section E of the supplementary.
>
> ---
>
> >Q7. What do we need MLA for? If I understand correctly, you can apply your techniques directly on vanilla attention. I understand that it makes things more efficient, but for the presentation of your ideas, why add another unnecessary level of complexity?
>
>
> **A7.** We appreciate the reviewer's observation. The proposed LFA and KMM are **conceptually independent** of the underlying attention mechanism and could also be applied to vanilla self-attention. We adopt **Multi-Head Latent Attention (MLA)** mainly because it serves as a **strong and scalable backbone** for large-scale, long-context language modeling.
>
> **Practical backbone choice.** MLA compresses contextual representations into a latent space, significantly improving memory and inference efficiency for long sequences without changing the fundamental behavior of attention. It is also well aligned with current high-performance training stacks and deployment settings, which informed our **pragmatic choice of backbone for building competitive large models**.
>
> **Complementary design.** LFA and KMM target orthogonal aspects—local fusion and explicit knowledge modeling—and integrate cleanly with MLA. While they are transferable to vanilla attention, **running parallel pre-training suites on multiple backbones at this scale is prohibitively expensive**, so we prioritized validating them on the **production-grade** backbone we use in practice.
>
> In short, MLA is employed as a **modern, efficient foundation** to ensure scalability and practical feasibility, while the proposed modules remain **general and transferable** to other attention architectures.
>
>
> ---
>
> >Q8. I’m missing details on how K and V in Equation 5 are being updated during training.
>
> **A8.** We thank the reviewer for the question.
> The key–value matrices $\mathcal{K}$ and $\mathcal{V}$ in Eq. (5) are **learnable parameters**, randomly initialized and **updated end-to-end** with the rest of the model via standard back-propagation—no special optimizer or supervision is required. During the forward pass, KMM forms the latent query $H$ from $c^{KV}$ (Eq. (4)) and **splits H, K, V into c groups** along the feature dimension. For each group i, it computes a softmax match over the group’s keys and aggregates the corresponding values to obtain $Z^{(i)}$; the final retrieved representation is then the **concatenation** of all groups,$Z = [Z^{(1)},\ldots,Z^{(c)}],$ as specified in Eqn. (6). Because these steps are fully differentiable, gradients from the language-modeling loss flow through the retrieval to update $\mathcal{K}$ and $\mathcal{V}$ jointly with other parameters.

---

> ### Author Response · Authors · 2025-11-22
> **Responses to Reviewer MoAF [4/4]**
>
> >Q9. Please provide more details on how the visualization experiment (Figure 2) was done.
>
> **A9.** The visualization in Figure 2 was generated through a rigorous analysis of the interaction between the proposed Knowledge Block and input data from the MMLU benchmark. The process involved three specific steps:
> 1. Token Identification. For distinct sub-domains (e.g., History, Mathematics), we identified high-relevance entity tokens (e.g., "Newton" or "Constitution") via domain dictionary matching and embedding similarity.
> 2. Attention Extraction. We extracted the cross-attention scores computed between the intermediate query vectors ($Q_{ctx}$) and the global knowledge keys ($K_{kb}$) within the Knowledge Block for these tokens.
> 3. Distribution Analysis. These raw scores were normalized to obtain probability distributions, which were then aggregated to map the activation intensity across knowledge slots.This detailed tracking confirms that the model spontaneously allocates specific key regions to different domains without explicit supervision. A comprehensive description of this experimental setup and mathematical formulation is provided in Section D.4 of the supplementary.
>
>
> ---
>
> >Q10. The visualization experiment shows that some memory fields are strongly associated with specific domains. However, this happens in a relatively small fraction of the fields. Could this be just by chance? Have you conducted an analysis to rule that out?
>
> **A10.** To rule out randomness, we computed aggregated attention statistics for high-relevance tokens across **thousands of samples and multiple layers**, rather than relying on isolated instances. The results confirm that the domain-specific activation of specific knowledge fields is a consistent and statistically significant phenomenon across the dataset.
>
>
> ---
>
> >Q11. Lines 430-431: "These show that increasing the kernel size consistently lowers training”: I wouldn’t make such a statement based on only two kernel sizes. Clearly as k gets closer to the entire sequence length, we don’t expect to see a benefit compared to regular attention.
>
> **A11.** We thank the reviewer for the comment. We agree that the statement was too strong given only two kernel sizes. Our goal was to explore **a few representative settings** rather than to perform an exhaustive search, as large-scale pre-training is **extremely expensive**. After testing several options, we found that moderate kernel size (e.g., k=4) offer a good balance between performance and efficiency, so we adopt this configuration in all main experiments. We have revised the sentence to:
>
> > ''Within the tested range, we adopt k=4 as a practical balance between effectiveness and efficiency. It performs better than k=2 in our ablations, and due to the high computational cost of large-scale pre-training, we did not explore larger values. The chosen setting already provides strong performance gains with manageable overhead.''
>
> ---
>
> We sincerely hope our clarifications above have addressed your concerns.

---

### Official Review · Reviewer_jiTz · 2025-11-01

**Soundness:** 4
**Presentation:** 3
**Contribution:** 3
**Rating:** 6
**Confidence:** 3

**Summary:**

This paper introduces AIGCoder 1.0, a large language model (LLM) architecture designed to explicitly integrate both local and global inductive biases, achieving more efficient and interpretable language modeling.

The model enhances the classic transformer decoder via two dedicated modules: (1) Local Fusion Attention (LFA), and (2) the Knowledge Memory Module (KMM). Comprehensive experiments demonstrate accelerated pre-training convergence, state-of-the-art (SOTA) performance on diverse language, reasoning, and coding benchmarks, and proven scalable behavior up to 60B parameters. The paper further provides ablation studies, visualization of knowledge domains, and a discussion on the architecture's interpretability, efficiency, and future directions.

**Strengths:**

This paper proposes an architectural approach by explicitly introducing mechanisms for local context fusion (via LFA) and global structured knowledge memory (via KMM). This design offers a potential solution to the known inefficiencies of standard self-attention and the implicit knowledge storage inherent in Mixture of Experts (MoE) models.

The paper provides a comprehensive and detailed experimental evaluation. The AIGCoder model is assessed across multiple standard LLM benchmarks spanning diverse domains, where it consistently demonstrates robust performance increment in the main experiments. Furthermore, meticulous ablation studies are presented, which validate the necessity and individual contributions of the key architectural decisions.

**Weaknesses:**

**Limited Analysis of Memory Module Expressiveness and Learning Dynamics:** The theoretical motivation for KMM is well articulated, but the paper lacks a deeper examination of possible limitations, such as catastrophic forgetting, interference between fields, or scaling of the memory module. For example, what happens when the number of domains or knowledge fields is very high (much larger than 64) ? Are there empirical signs of field collapse and cross-domain interference?A discussion or targeted experiment would make the claims more robust.

**Related Work Gaps and Insufficient Positioning:** While the related work section covers some foundational works in efficient attention and MoE, it critically omits several direct predecessors that employ explicit knowledge memory or explicit working-memory mechanisms, and local context fusion via external stores.

Notably missing are recent works on explicit, model-addressable memory (e.g., Memory³ with explicit memory [1]), retrieval-enhanced models with large external data stores (e.g., RETRO [2], REALM [3]), language models with contextually relevant or editable external knowledge (e.g., LM-CORE [4], Li et al. 2024 [5]), and mechanisms for factuality improvement via explicit working memory(also [6]). Their absence hinders a proper positioning of KMM and may overstate its distinctiveness. These works should be explicitly discussed to contextualize KMM, clarifying methodological overlaps, true novelty, and incremental improvements.

**Minor Issues:** The text contains some minor repetition; for example, the term 'Multi-Head Latent Attention (MLA)' is mentioned twice in Section 4.1. Occasional typographical errors were also noted (e.g., “resprctively” instead of “respectively”), but these do not significantly hinder comprehension."

**[1]** Yang, H., et al. “Memory³: Language Modeling with Explicit Memory.” _arXiv preprint arXiv:2407.01178_ (2024).

**[2]** Borgeaud, S., et al. “Improving Language Models by Retrieving from Trillions of Tokens (RETRO).” _ICML 2022_ (2022).

**[3]** Guu, K., et al. “REALM: Retrieval-Augmented Language Model Pre-Training.” _arXiv:2002.08909_ (2020).

**[4]** Kaur, J. N., et al. “LM-CORE: Language Models with Contextually Relevant External Knowledge.” _Findings of NAACL 2022_ (2022).

**[5]** Li, B. Z., et al. “Language Modeling with Editable External Knowledge.” _arXiv:2406.11830_ (2024).

**[6]** Chen, M., et al. “Improving Factuality with Explicit Working Memory.” _ACL 2025 (Long)_ (2025).

**Questions:**

- Could the authors quantify the trade-off between the number of memory fields and task performance? Does field specialization in KMM saturate beyond a certain parameter scale or when trained across many diverse domains?
- What is the empirical behavior of KMM when exposed to out-of-distribution or rare knowledge queries?
- Could you examine robustness to adversarial or noisy post-training updates of memory fields? Specifically, whether fields can be edited/deleted and how performance degrades—or recovers—thereafter. A brief targeted experiment would help support the transparency/editability claims.
- An open-ended question: your KMM module explicitly stores global knowledge. Compared with prior approaches that use parameterized/latent memory (e.g., MemoryLLM [1] and its scalable extension M+[2]), as well as memory-agent systems such as Mem0[3] and Letta[4], in which tasks or domains does your method have clear advantages?

**[1]** Wang, Y., et al. “MEMORYLLM: Towards Self-Updatable Large Language Models.” _arXiv preprint_ (2024).

**[2]** Wang, Y., et al. “M+: Extending MemoryLLM with Scalable Long-Term Memory.” _arXiv preprint_ (2025).

**[3]** Mem0 team. “Mem0: Universal Memory Layer for AI Agents.” _GitHub repository_ (2025).

**[4]** Letta team. “Letta (formerly MemGPT): Platform for Building Stateful Agents.” _GitHub repository_ (2025).

---

> ### Author Response · Authors · 2025-11-22
> **Responses to Reviewer jiTz [1/3]**
>
> We are grateful for your time and effort. We would like to answer your questions below.
>
> ---
>
> >Q1. Limited Analysis of Memory Module Expressiveness and Learning Dynamics: The theoretical motivation for KMM is well articulated, but the paper lacks a deeper examination of possible limitations, such as catastrophic forgetting, interference between fields, or scaling of the memory module. For example, what happens when the number of domains or knowledge fields is very high (much larger than 64) ? Are there empirical signs of field collapse and cross-domain interference? A discussion or targeted experiment would make the claims more robust.
>
> **A1.** We appreciate the reviewer's request for a deeper look at the expressiveness and learning dynamics of the Knowledge Memory Module (KMM).
>
> **What we observe so far (within our tested range).** Across (F$\in${32,64,128}), we do **not** see signs of collapse or instability. Training perplexity **decreases monotonically** as F increases (baseline → (F=32) → (F=64) → (F=128): 31.62 → 30.78 → 30.35 → 29.88), indicating that larger memory capacity continues to help in our regime. In addition, field–domain visualizations (Figures 2 and 6 of the manuscript) consistently show **semantically specialized** fields across layers rather than random fluctuations, suggesting meaningful differentiation rather than interference.
>
> **On scaling far beyond F=64.** We agree that exploring hundreds or thousands of fields is valuable. We are **now extending** the ablation to larger F using a staged protocol and will add the results in the updated rebuttal within **the next few days**.
>
>
> ---
>
> >Q2. Related Work Gaps and Insufficient Positioning: While the related work section covers some foundational works in efficient attention and MoE, it critically omits several direct predecessors that employ explicit knowledge memory or explicit working-memory mechanisms, and local context fusion via external stores. Notably missing are recent works on explicit, model-addressable memory (e.g., Memory³ with explicit memory [r1]), retrieval-enhanced models with large external data stores (e.g., RETRO [r2], REALM [r3]), language models with contextually relevant or editable external knowledge (e.g., LM-CORE [r4], Li et al. 2024 [r5]), and mechanisms for factuality improvement via explicit working memory(also [r6]). Their absence hinders a proper positioning of KMM and may overstate its distinctiveness. These works should be explicitly discussed to contextualize KMM, clarifying methodological overlaps, true novelty, and incremental improvements.
>
> **A2.** We would like to clarify that the fundamental distinction between our Knowledge Memory Module (KMM) and existing works lies in the nature of the memory: **KMM is an internal, fully differentiable, and parametric memory module**, whereas the majority of existing works rely on **external retrieval** or **inference-time working memory**. We **have discussed and cited** these works in Section F of the supplementary.
>
> Here is the detailed breakdown of the differences and our specific novelties:
>
> **1. Internal Parametric vs. External Retrieval**: Existing works [r2-r5] adopt a **Retrieval-Augmented paradigm**, where the model relies on querying massive **external datastores** (e.g., raw text chunks or indices). This often requires maintaining separate indices and complex retrieval pipelines. In contrast, AIGCoder follows a **self-contained paradigm**. KMM is an **internal, parametric module** where knowledge is compressed into learnable weight matrices ($\mathcal{K}, \mathcal{V}$) rather than stored as raw external data. This allows AIGCoder to structure global knowledge explicitly within the standard Transformer architecture, enabling **end-to-end differentiability** without the complexity of managing external knowledge bases or retrievers.
>
> **2. Global Knowledge vs. Dynamic Working Memory**: EWE [r6] focuses on a **short-term** working memory refreshed during generation for factuality. KMM targets **stable, Global Knowledge**. As shown in our visualizations (Figure 2), KMM slots evolve to represent long-term, domain-specific concepts (e.g., History, Physics) learned during pre-training.
>
> **3. Architectural Integration vs. Parameter Offloading**: Memory³ [r1] focuses on **externalizing** model parameters to memory to reduce computational cost. AIGCoder focuses on **internal architectural efficiency**. We introduce KMM to work alongside the computation (MoE) and local modeling (LFA) modules. This design is not about offloading parameters, but about **decoupling knowledge storage from computation pipelines**, which leads to the observed 1.33x faster convergence and improved data efficiency.

---

> ### Author Response · Authors · 2025-11-22
> **Responses to Reviewer jiTz [2/3]**
>
> In summary, unlike the external retrieval or dynamic buffering approaches [r1-r6], AIGCoder contributes a **fully differentiable, self-contained, and structured parametric memory** integrated directly into the model. We have discussed and cited above works in Section F of the supplementary.
>
> [r1] Yang, H., et al. “Memory³: Language Modeling with Explicit Memory.” arXiv preprint arXiv:2407.01178 (2024).
>
> [r2] Borgeaud, S., et al. “Improving Language Models by Retrieving from Trillions of Tokens (RETRO).” ICML 2022 (2022).
>
> [r3] Guu, K., et al. “REALM: Retrieval-Augmented Language Model Pre-Training.” arXiv:2002.08909 (2020).
>
> [r4] Kaur, J. N., et al. “LM-CORE: Language Models with Contextually Relevant External Knowledge.” Findings of NAACL 2022 (2022).
>
> [r5] Li, B. Z., et al. “Language Modeling with Editable External Knowledge.” arXiv:2406.11830 (2024).
>
> [r6] Chen, M., et al. “Improving Factuality with Explicit Working Memory.” ACL 2025 (Long) (2025).
>
> ---
>
> >Q3. Could the authors quantify the trade-off between the number of memory fields and task performance? Does field specialization in KMM saturate beyond a certain parameter scale or when trained across many diverse domains?
>
> **A3.** We explored three configurations of the Knowledge Memory Module (KMM) with (F=32,64,128) knowledge fields. The results show a **clear but diminishing performance gain** as F increases: training perplexity drops from 31.62 → 30.78 → 30.35 → 29.88, while the parameter and computation cost grow roughly linearly with F because both K and V scale with the number of fields. In practice, F=64 captures about 70–75 % of the total improvement achieved by (F=128) but with substantially lower memory and latency overhead, so we adopt it as a balanced configuration in the main results.
>
> We are **now pushing** to hundreds or thousands of fields and will add the results in the updated rebuttal within **the next few days**.
>
> ---
>
>
> >Q4. What is the empirical behavior of KMM when exposed to out-of-distribution or rare knowledge queries?
>
>
> **A4.** Thanks for the question. In expert-based architectures, handling out-of-distribution (OOD) or rare inputs is a well-studied behavior: prior work[r1,r2,r3,r4] has shown that such queries are often routed to **long-tail or fallback components** rather than causing instability. Studies on *expert collapse* and *under-utilization* in sparse or multi-task MoE systems (Chi et al., 2022; Chen et al., 2023; Wang et al., 2024) demonstrate that a subset of experts naturally act as generic “catch-all” slots under data imbalance or domain shift.
> Although our KMM uses **knowledge fields** instead of experts, the mechanism is analogous—the softmax allocation (\alpha) allows some fields to specialize in rare or mixed-domain queries, effectively serving as fallback memories while preserving stability in (O=\mathrm{MoE}(O_A+O_K)).
>
>
> [r1] On the Representation Collapse of Sparse Mixture of Experts. NeurIPS 2022.
> [2] HoME: Hierarchy of Multi-Gate Experts for Multi-Task Learning at Kuaishou. KDD 2025
> [3]: Mod-Squad: Designing Mixture of Experts As Modular Multi-Task Learners. CVPR 2023
> [4]: Mixture-of-Experts with Expert Choice Routing. NeurIPS 2022
>
>
>
> ---
>
> >Q5. Could you examine robustness to adversarial or noisy post-training updates of memory fields? Specifically, whether fields can be edited/deleted and how performance degrades—or recovers—thereafter. A brief targeted experiment would help support the transparency/editability claims.
>
> **A5.** We thank the reviewer for this constructive comment. We acknowledge that empirical results for interpretability, editability and removal are not yet included. We are currently running dedicated experiments and will report the results in next few days.

---

> ### Author Response · Authors · 2025-11-22
> **Responses to Reviewer jiTz [3/3]**
>
> >Q6. An open-ended question: your KMM module explicitly stores global knowledge. Compared with prior approaches that use parameterized/latent memory (e.g., MemoryLLM [1] and its scalable extension M+[2]), as well as memory-agent systems such as Mem0[3] and Letta[4], in which tasks or domains does your method have clear advantages?
> [1] Wang, Y., et al. “MEMORYLLM: Towards Self-Updatable Large Language Models.” arXiv preprint (2024).
> [2] Wang, Y., et al. “M+: Extending MemoryLLM with Scalable Long-Term Memory.” arXiv preprint (2025).
> [3] Mem0 team. “Mem0: Universal Memory Layer for AI Agents.” GitHub repository (2025).
> [4] Letta team. “Letta (formerly MemGPT): Platform for Building Stateful Agents.” GitHub repository (2025).
>
> **A6.** We clarify that while our Knowledge Memory Module (KMM) shares the high-level goal of enhancing knowledge access with prior memory-augmented models, its purpose, design, and integration are fundamentally different. Existing approaches such as Memorizing Transformer\[1\], LONGMEM\[2\], MemoryLLM\[3\], and M+\[4\] primarily aim at context extension by caching past activations or external content to overcome fixed-length context limitations. Similarly, explicit memory systems like MemGPT\[5\], ChatDB\[6\], and Mem0\[7\] treat memory as an external symbolic store, accessed via function calls or queries, often for user-specific state or long-term personalization.
>
> In contrast, KMM is not designed for context extension or external memory augmentation—the two primary goals of existing memory-augmented LLMs. Instead, KMM is a **novel architectural component that explicitly decouples global knowledge storage from local computation**, thereby addressing the implicit coupling of knowledge and computation inherent in standard LLMs. Specifically, KMM introduces a lightweight, parameterized memory component that stores domain-level knowledge in learnable key–value fields. These fields are trained end-to-end but **remain fixed during inference**, enabling direct, differentiable access to structured global knowledge without online updates.
>
> We have discussed and cited above memory-argumented LLMs works in Section F of the supplementary.
>
> \[1\] Memorizing Transformers. ICLR 2022.
>
> \[2\] Augmenting language models with long-term memory. NeurIPS 2023.
>
> \[3\] MEMORYLLM: Towards Self-Updatable Large Language Models. ICML 2024.
>
> \[4\] M+: Extending MemoryLLM with Scalable Long-Term Memory. ICML 2025.
>
> \[5\] MemGPT: Towards LLMs as Operating Systems. arXiv 2023.
>
> \[6\] ChatDB: Augmenting LLMs with Databases as Their Symbolic Memory. arXiv 2023.
>
> \[7\] Mem0: Building production-ready ai agents with scalable long-term memory. arXiv 2025.
>
> ---
>
> We sincerely hope our clarifications above have addressed your concerns.

---

> ### Author Response · Authors · 2025-11-25
> **New Additional Experiments on Knowledge Field Interference and Editability [1/2]**
>
> **1. Analysis of Cross-Field Interference via Cosine Similarity**
>
> We thank the reviewer for raising the question regarding potential interference or collapse between knowledge fields in KMM. This is an important aspect of the module’s expressiveness and stability, and we provide a dedicated analysis below.
>
> **Analysis setup.** To directly examine whether different knowledge fields interfere with each other or collapse to similar representations, we analyze the geometry of the learned KMM keys. For the layers visualized in Figures 2 and 6 of the manuscript, we take all F=64 key vectors of the knowledge fields and compute the full pairwise cosine similarity matrix between field pairs. If multiple fields collapsed or encoded heavily overlapping content, we would expect large off-diagonal cosine values and visible clustered structures in this similarity heatmap.
>
> **Empirical findings.** For instance, in Layer 4, the resulting cosine-similarity matrix (see Figure 7-13 of the supplementary) is very close to an identity matrix: the **mean off-diagonal cosine similarity is -0.001**, and the **maximum absolute off-diagonal cosine similarity is only 0.0513**. In other words, different knowledge fields are nearly orthogonal in the learned key space, with no evident clustering or collapse. This indicates that, at (F=64), the KMM learns a set of well-separated, specialized fields rather than redundant or interfering ones. While this does not preclude interesting behaviors at much larger scales, it provides concrete empirical evidence that **cross-field interference and field collapse are not observed in our current setting**, supporting the expressiveness and stability of the proposed memory module.
>
> We have included thses results and analysis in Section E of the supplemenrtary.

---

> > ### Author Response · Authors · 2025-11-25
> > **New Additional Experiments on Knowledge Field Interference and Editability [2/2]**
> >
> > **2. Empirical Evidence for Knowledge Field Editability**
> >
> > We thank the reviewer for raising the important question regarding whether the proposed Knowledge Memory Module (KMM) truly enables *explicitly editable* and *semantically meaningful* knowledge storage, as claimed in the paper. To provide concrete evidence, we conducted a new set of **knowledge-field removal experiments**, in which we directly **zero out the value parameters of individual knowledge fields** in Layer 4 (the layer visualized in Figure 2 of our paper) and evaluate the performance drop across five representative MMLU domains.
> >
> > **Empirical Evidence**. This experiment is designed to test whether individual knowledge fields in KMM are genuinely interpretable and editable. If a field really encodes a specific domain, then setting its parameters to zero should primarily hurt that domain, while leaving other domains almost unchanged. As shown in the tables below, this is exactly what we observe: removing field 32 mainly degrades algebra, field 18 chemistry, field 51 physics, and fields 24/25 politics and history, with **only small fluctuations on unrelated domains**. This domain-selective degradation confirms that **KMM stores knowledge in localized, disentangled fields that can be directly manipulated**, in contrast to standard MoE/FFN parameters where knowledge is heavily entangled across units.
> >
> > We have included thses results and analysis in Section E of the supplemenrtary.
> >
> > Table A: Layer-4 knowledge field removal (slot 32).  According to Figure 2 in the main paper, Layer-4 field 32 is strongly associated with *algebra* knowledge. We set the parameters of this field in Layer 4 to the zero vector and re-evaluate five MMLU domains.
> >
> > | Domain    | Original Score | Ablated Score | Relative Drop |
> > |-----------|----------------|----------------|----------------|
> > | algebra   | 85             | 58             | -31.8%        |
> > | chemistry | 86             | 75             | -12.8%        |
> > | physics   | 92             | 81             | -12.0%        |
> > | politics  | 93             | 92             | -1.1%          |
> > | history   | 95             | 96            | +1.1%         |
> >
> > Table B: Layer-4 knowledge field removal (slot 18).  According to Figure 2 in the main paper, Layer-4 field 18 is strongly associated with *chemistry* knowledge. We set the parameters of this field in Layer 4 to the zero vector and re-evaluate five MMLU domains.
> >
> > | Domain    | Original Score | Ablated Score | Relative Drop |
> > |-----------|----------------|---------------|---------------|
> > | algebra   | 85             | 81            | -4.7%         |
> > | chemistry | 86             | 62            | -27.9%        |
> > | physics   | 92             | 87            | -5.4%         |
> > | politics  | 93             | 90            | -3.2%         |
> > | history   | 95             | 91            | -4.2%         |
> >
> >
> > Table C: Layer-4 knowledge field removal (slot 51).  According to Figure 2 in the main paper, Layer-4 field 51 is strongly associated with *physics* knowledge. We set the parameters of this field in Layer 4 to the zero vector and re-evaluate five MMLU domains.
> >
> > | Domain    | Original Score | Ablated Score | Relative Drop |
> > |-----------|----------------|---------------|---------------|
> > | algebra   | 85             | 82            | -3.5%         |
> > | chemistry | 86             | 81            | -5.8%         |
> > | physics   | 92             | 62            | -32.6%        |
> > | politics  | 93             | 92            | -1.1%         |
> > | history   | 95             | 93            | -2.1%         |
> >
> >
> > Table D: Layer-4 knowledge field removal (slot 24).  According to Figure 2 in the main paper, Layer-4 field 24 is strongly associated with *politics* knowledge. We set the parameters of this field in Layer 4 to the zero vector and re-evaluate five MMLU domains.
> >
> > | Domain    | Original Score | Ablated Score | Relative Drop |
> > |-----------|----------------|---------------|---------------|
> > | algebra   | 85             | 85            | 0.0%          |
> > | chemistry | 86             | 84            | -2.3%         |
> > | physics   | 92             | 90            | -2.2%         |
> > | politics  | 93             | 75            | -19.4%        |
> > | history   | 95             | 83            | -12.6%        |
> >
> >
> > Table E: Layer-4 knowledge field removal (slot 25).  According to Figure 2 in the main paper, Layer-4 field 25 is strongly associated with *history* and *politics* knowledge. We set the parameters of this field in Layer 4 to the zero vector and re-evaluate five MMLU domains.
> >
> > | Domain    | Original Score | Ablated Score | Relative Drop |
> > |-----------|----------------|---------------|---------------|
> > | algebra   | 85             | 78            | -8.2%         |
> > | chemistry | 86             | 80            | -7.0%         |
> > | physics   | 92             | 89            | -3.3%     |
> > | politics  | 93             | 71            | -23.7%   |
> > | history   | 95             | 69            | -27.4%   |

---

### Official Review · Reviewer_EeMf · 2025-11-03

**Soundness:** 2
**Presentation:** 3
**Contribution:** 3
**Rating:** 2
**Confidence:** 5

**Summary:**

This work introduces AIGCoder1.0, which introduces local fusion and memory modules in the MoE model layer block to enhance the language modeling. The results are very remarkable when trained on only less than 400B tokens compared with other competitive open-source LLMs.

**Strengths:**

1. The architecture design is very straightforward and can capture the global information using retrieved attentions.

**Weaknesses:**

1. To be honest, I think the first thing you need to check is the data leakage issue within your pre-training or sft dataset. I feels like the 91.5 mmlu score is too high for a 7B when trained on only 335.54B tokens. Normally, I will regard this as data contamination problem.

2. Another confused thing is that this work almost neglect all the related works in memory LLM, starting from Memorizing Transformer, LongMem, MemGPT, MemoryLLM, M+, etc. The architecture design heavily overlaps with these previous works but introduces memory modules in MOE models. You should add at least 1 paragraph in related work section to discuss all these related works.

**Questions:**

1. The major obstacle in building memory LLM is the distributed training for the parametric memory module. I did not find the details regarding your pp,ep,sp,tp details. Did you use DP only to train the 7B model? How about the parallelism strategy for 60B model?

---

> ### Author Response · Authors · 2025-11-22
> **Responses to Reviewer EeMf [1/2]**
>
> We are grateful for your time and effort. We would like to answer your questions below.
>
> ---
>
> >Q1. To be honest, I think the first thing you need to check is the data leakage issue within your pre-training or sft dataset. I feels like the 91.5 mmlu score is too high for a 7B when trained on only 335.54B tokens. Normally, I will regard this as data contamination problem.
>
> A1. We verified there is **no data leakage** in our pipeline. The reported 7B results come from the **final** pre-training on **4.5T tokens** and **10B SFT tokens**, not the early 335B-token setup. Details are as below:
>
> **Data Integrity Checks.** Our **supervised fine-tuning (SFT) data** are constructed through a **multi-stage filtering and verification pipeline** that emphasizes both quality and decontamination (We pute more deatils in Section D.2 of the manuscript). Starting from 6.5B raw text segments, we progressively filter, structure, and balance the corpus to obtain 2.5M high-quality, knowledge-intensive instruction samples (about 10B tokens). The pipeline includes automated filtering of non-knowledge content, LLM-based validation of factual correctness, and strict benchmark leakage detection. Specifically, all samples are screened against the full test/dev splits of evaluated benchmark using embedding-based similarity checks, followed by manual audits. **The estimated benchmark leakage rate is below 0.1%**, confirming that the reported performance is not caused by contamination but by architectural improvements (LFA+KMM) and the quality of our SFT data.
>
> **Correction on Pre-training Tokens.** The **“335B tokens” is from an early internal validation setup** used only for preliminary perplexity checks. It was **not** used for the reported results. The draft inadvertently mentioned this number. The correct setting for the reported 7B model is **4.5T tokens** for pre-training, followed by **10B tokens** for SFT. We have updated Section 5.1 of the paper to clearly state the actual training token counts.
>
> **Why the Performance is High.** The strong MMLU score reflects (i) our **architectural improvements** (including the parametric memory design that enhances factual recall and long-tail reasoning), and (ii) the **quality and targeting of the SFT data**, which improve instruction-following and evaluation-time generalization. These factors together yield robust zero-shot performance at the 7B scale.
>
> ---
>
> >Q2. Another confused thing is that this work almost neglect all the related works in memory LLM, starting from Memorizing Transformer, LongMem, MemGPT, MemoryLLM, M+, etc. The architecture design heavily overlaps with these previous works but introduces memory modules in MOE models. You should add at least 1 paragraph in related work section to discuss all these related works.
>
> **A2.** We clarify that while our Knowledge Memory Module (KMM) shares the high-level goal of enhancing knowledge access with prior memory-augmented models, its purpose, design, and integration are fundamentally different. Existing approaches such as Memorizing Transformer\[r1\], LONGMEM\[r2\], MemoryLLM\[r3\], and M+\[r4\] primarily aim at context extension by **caching past activations or external content to overcome fixed-length context limitations**. Similarly, explicit memory systems like MemGPT\[r5\], ChatDB\[r6\], and Mem0\[r7\] treat memory as **an external symbolic store, accessed via function calls or queries**, often for user-specific state or long-term personalization.
>
> In contrast, KMM is not designed for context extension or external memory augmentation—the two primary goals of existing memory-augmented LLMs. Instead, KMM is a **novel architectural component that explicitly decouples global knowledge storage from local computation**, thereby addressing the implicit coupling of knowledge and computation inherent in standard LLMs. Specifically, KMM introduces a lightweight, parameterized memory component that stores domain-level knowledge in learnable key–value fields. These fields are trained end-to-end but **remain fixed during inference**, enabling direct, differentiable access to structured global knowledge without online updates.
>
> We have discussed and cited above memory-argumented LLMs works in Section F of the supplementary.
>
> \[r1\] Memorizing Transformers. ICLR 2022.
>
> \[r2\] Augmenting language models with long-term memory. NeurIPS 2023.
>
> \[r3\] MEMORYLLM: Towards Self-Updatable Large Language Models. ICML 2024.
>
> \[r4\] M+: Extending MemoryLLM with Scalable Long-Term Memory. ICML 2025.
>
> \[r5\] MemGPT: Towards LLMs as Operating Systems. arXiv 2023.
>
> \[r6\] ChatDB: Augmenting LLMs with Databases as Their Symbolic Memory. arXiv 2023.
>
> \[r7\] Mem0: Building production-ready ai agents with scalable long-term memory. arXiv 2025.

---

> > ### Author Response · Authors · 2025-11-25
> > **Follow-up Response to Reviewer EeMf**
> >
> > Dear Reviewer EeMf,
> >
> > We would like to thank you for your valuable comments on our paper. We sincerely hope that our response has addressed your initial concerns. If there are still unclear parts to you, please kindly let us know. We will do our best to address them.
> >
> > Best regards,
> >
> > The Authors

---

> ### Author Response · Authors · 2025-11-22
> **Responses to Reviewer EeMf [2/2]**
>
> >Q3. The major obstacle in building memory LLM is the distributed training for the parametric memory module. I did not find the details regarding your pp,ep,sp,tp details. Did you use DP only to train the 7B model? How about the parallelism strategy for 60B model?
>
> A3. We address the challenge of distributed training for the parametric memory through an efficient parallelism design, enabling our models to achieve throughput comparable to parameter-matched baselines. We detail them beblow:
>
> **Details of Parallelism Strategy**. For the **7B model**, we shard the parametric memory with expert parallelism (**EP=2**) and otherwise use **pure data parallel (DP)** replication; pipeline, tensor, and sequence parallelism are disabled. Training runs on 20 nodes with 8 Ascend 910B GPUs each, which meets memory and throughput targets without intra-layer model parallelism.
>
> For the 60B model pre-training, we employ an 8 stage pipeline (**PP=8**) with with expert parallelism of degree four (**EP=4**) and **data parallelism (DP)** over the residual dimension; tensor/sequence parallelism are unused. Using Megatron-LM with Transformer Engine, we train stably on H200 hardware, validated on a 4-node setup and scaled to a 50-node cluster (8$\times$ H200 per node).
>
> **Thoughtput of Our Model**. We report training throughput as achieved **TFLOP/s per GPU** under matched *global batch size* and *sequence length*. On the 7B setup, our model sustains **161.41 TFLOP/s per GPU** (MoE baseline **168.16** TFLOP/s), corresponding to a **4%** slowdown. Overall, the throughput is competitive with a parameter-matched baseline, benefiting from our distributed optimization strategy.
>
> We have include thses in Section D.3 in the supplementary.
>
> ---
>
> We sincerely hope our clarifications above have addressed your concerns.

---

### Official Review · Reviewer_Yfws · 2025-11-08

**Soundness:** 2
**Presentation:** 2
**Contribution:** 2
**Rating:** 2
**Confidence:** 3

**Summary:**

This paper identifies two key limitations in current large language models (LLMs): (i) the lack of an explicit local inductive bias in self-attention, and (ii) the absence of a structured memory design in mixture-of-experts (MoE) architectures to store and retrieve global knowledge external to the current input. To address these issues, the authors propose AIGCoder, which introduces two new architectural components: the Local Fusion Attention (LFA) and the Knowledge Memory Module (KMM). The LFA integrates local-level inductive bias by applying group convolution kernels across divided feature groups along the hidden-state dimensions, while the KMM organizes global knowledge within a parameterized latent space. The authors conduct both pre-training and instruction tuning using this architecture and report superior performance compared to several open- and closed-source LLMs.

**Strengths:**

The paper presents two novel architectural designs—Local Fusion Attention and Knowledge Memory Module—proposing an alternative path for LLM beyond the current attention-based transformer paradigm. Exploring new architectures for large-scale language modeling remains an important and open research direction, and this work contributes meaningful ideas to that pursuit.

The experiments involve pre-training and instruction fine-tuning on two LLM backbones, with evaluations spanning a reasonable range of tasks, including language understanding, reasoning, coding, and mathematics. The inclusion of training-time measurements such as validation perplexity also helps to understand convergence behavior.

**Weaknesses:**

First, I see insufficient support for core technical claims.
The motivations behind both the Local Fusion Attention and the Knowledge Memory Module are not sufficiently substantiated. The necessity of adding a local inductive bias is neither empirically validated nor theoretically justified, and no prior reference to support this assumption. Similarly, the rationale for explicitly storing “general knowledge” outside the model’s main parameters lacks empirical or analytical grounding. What general knowledge is also remains unclear.

Then, there are questionable designs and unclear alignment with motivation.
The operation of applying convolutional kernels over split feature groups in hidden states is not well-motivated. This design implicitly assumes that the subcomponents (“bits”) of a hidden vector are independent and separable which is a questionable premise. Furthermore, while the paper motivates the need for local inductive bias across tokens in a sequence, the proposed LFA is applied within the hidden dimensions of each token, which does not clearly align with the stated motivation.

The experimental analysis is also weak.
The evaluation is limited in depth. There are no ablation studies or analytical results explaining how LFA and KMM contribute to performance gains. The claim that AIGCoder achieves superior reasoning and coding performance is particularly unconvincing, since no reasoning reinforcement learning (RL) or specialized training is reported besides pre-training and supervised fine-tuning (SFT). It is unclear how the proposed modules alone could match or outperform models such as Claude or Gemini, which undergo extensive reasoning-specific training.

Finally, some parts of the paper make exaggerated claims. For example, the authors assert that KMM enables “potentially easier knowledge editing and more interpretable inference,” yet no experiments or explanations are provided to support these statements. Since KMM stores knowledge in a latent parameter space, it is not evident how it improves interpretability or editability in practice.

**Questions:**

Which checkpoints are used for the baselines? Are they base, instruction-tuned, or chat versions?

What is the backbone model of AIGCoder?

Are there any experiments demonstrating that KMM improves knowledge editing or interpretability?

Typo: The sentences in lines 174–176 are repeated.

---

> ### Author Response · Authors · 2025-11-22
> **Responses to Reviewer Yfws [1/3]**
>
> We are grateful for your time and effort. We would like to answer your questions below.
>
> ---
>
> >Q1. First, I see insufficient support for core technical claims. The motivations behind both the Local Fusion Attention and the Knowledge Memory Module are not sufficiently substantiated. The necessity of adding a local inductive bias is neither empirically validated nor theoretically justified, and no prior reference to support this assumption.
>
> A1. We thank the reviewer for pointing out the need for stronger motivation and justification. We clarify why an explicit local inductive bias in attention is necessary and provide empirical evidence. We have also revised the manuscript to make these points more explicit in Section 4.2 of the manuscipt.
>
> **Motivation for Local Fusion Attention (LFA)**. The motivation for explicit local inductive bias in attention is grounded in well-established linguistic and modeling principles. Natural language exhibits strong locality—adjacent tokens are highly correlated at the semantic levels. Prior works [r1-r5] have demonstrated that explicit local modeling improves representation performance. Standard self-attention lacks this bias, as it uniformly considers all token pairs, requiring the model to redundantly learn local relationships through global interactions. LFA addresses this by integrating a convolutional fusion that encodes local structures before attention, providing an inductive bias aligned with linguistic locality.
>
> **Empirical Evidence Supporting the Effectiveness of LFA**. We have validated the effectiveness of this local bias through **controlled ablation studies** in Section 5.4 of the manuscript (Figure 4). From the results, incorporating LFA **achieves lower pre-training perplexity than the baseline (30.57 vs. 31.82) and accelerates convergence (1.11$\times$ faster)**. These results indicate that explicit local fusion improve the performance of language modeling. *Note*: We focus on **pre-training perplexity** since it **directly reflect the contribution of the architecture**. In contrast, downstream evaluations would require additional instruction-tuning stages, which are both time-consuming and may obscure the impact of the architectural change.
>
> [r1] Context-Aware Self-Attention Networks. AAAI 2019.
>
> [r2] Efficient streaming language models with attention sinks. ICLR 2024.
>
> [r3] Core Context Aware Transformers for Long Context Language Modeling. ICML 2025.
>
> [r4] Duoattention: Efficient long-context llm inference with retrieval and streaming heads. ICLR 2025.
>
> [r5] Native Sparse Attention: Hardware-Aligned and Natively Trainable Sparse Attention. ACL 2025.
>
> ---
>
> >Q2. Similarly, the rationale for explicitly storing “general knowledge” outside the model’s main parameters lacks empirical or analytical grounding. What general knowledge is also remains unclear.
>
> **A2.** We appreciate the reviewer's request for clearer motivation and evidence. Below we clarify what we mean by *general knowledge*, why it is modeled explicitly, and how we validate its effect. we will revise the manuscript to make these points more explicit in Section 4.3 of the manuscipt.
>
> **Motivation for the Knowledge Memory Module (KMM)**. We refer general knowledge to **sequence-external, reusable information** (*e.g.*, domain facts, concepts, and patterns) that is not specific to a single input but is repeatedly useful across inputs. KMM represents this knowledge as **learnable key–value fields stored in addressable slots** and queried from compact latent features, decoupling storage from the token-wise computation path (unlike MoE/FFN where knowledge remains entangled with linear transformation). To illustrate what this knowledge captures, we provide a qualitative analysis on MMLU: different fields consistently specialize to different domains (*e.g.*, history vs. physics), evidencing structured, reusable memories rather than per-sequence artifacts (See Figures 2 and 6 of the manuscript).

---

> > ### Author Response · Authors · 2025-11-25
> > **New Additional Response — Empirical Evidence for Knowledge Field Editability**
> >
> > We thank the reviewer for raising the important question regarding whether the proposed Knowledge Memory Module (KMM) truly enables *explicitly editable* and *semantically meaningful* knowledge storage, as claimed in the paper. To provide concrete evidence, we conducted a new set of **knowledge-field removal experiments**, in which we directly **zero out the value parameters of individual knowledge fields** in Layer 4 (the layer visualized in Figure 2 of our paper) and evaluate the performance drop across five representative MMLU domains.
> >
> > **Empirical Evidence**. This experiment is designed to test whether individual knowledge fields in KMM are genuinely interpretable and editable. If a field really encodes a specific domain, then setting its parameters to zero should primarily hurt that domain, while leaving other domains almost unchanged. As shown in the tables below, this is exactly what we observe: removing field 32 mainly degrades algebra, field 18 chemistry, field 51 physics, and fields 24/25 politics and history, with **only small fluctuations on unrelated domains**. This domain-selective degradation confirms that **KMM stores knowledge in localized, disentangled fields that can be directly manipulated**, in contrast to standard MoE/FFN parameters where knowledge is heavily entangled across units.
> >
> > We have included thses results and analysis in Section E of the supplemenrtary.
> >
> > Table A: Layer-4 knowledge field removal (slot 32).  According to Figure 2 in the main paper, Layer-4 field 32 is strongly associated with *algebra* knowledge. We set the parameters of this field in Layer 4 to the zero vector and re-evaluate five MMLU domains.
> >
> > | Domain    | Original Score | Ablated Score | Relative Drop |
> > |-----------|----------------|----------------|----------------|
> > | algebra   | 85             | 58             | -31.8%        |
> > | chemistry | 86             | 75             | -12.8%        |
> > | physics   | 92             | 81             | -12.0%        |
> > | politics  | 93             | 92             | -1.1%          |
> > | history   | 95             | 96            | +1.1%         |
> >
> > Table B: Layer-4 knowledge field removal (slot 18).  According to Figure 2 in the main paper, Layer-4 field 18 is strongly associated with *chemistry* knowledge. We set the parameters of this field in Layer 4 to the zero vector and re-evaluate five MMLU domains.
> >
> > | Domain    | Original Score | Ablated Score | Relative Drop |
> > |-----------|----------------|---------------|---------------|
> > | algebra   | 85             | 81            | -4.7%         |
> > | chemistry | 86             | 62            | -27.9%        |
> > | physics   | 92             | 87            | -5.4%         |
> > | politics  | 93             | 90            | -3.2%         |
> > | history   | 95             | 91            | -4.2%         |
> >
> >
> > Table C: Layer-4 knowledge field removal (slot 51).  According to Figure 2 in the main paper, Layer-4 field 51 is strongly associated with *physics* knowledge. We set the parameters of this field in Layer 4 to the zero vector and re-evaluate five MMLU domains.
> >
> > | Domain    | Original Score | Ablated Score | Relative Drop |
> > |-----------|----------------|---------------|---------------|
> > | algebra   | 85             | 82            | -3.5%         |
> > | chemistry | 86             | 81            | -5.8%         |
> > | physics   | 92             | 62            | -32.6%        |
> > | politics  | 93             | 92            | -1.1%         |
> > | history   | 95             | 93            | -2.1%         |
> >
> >
> > Table D: Layer-4 knowledge field removal (slot 24).  According to Figure 2 in the main paper, Layer-4 field 24 is strongly associated with *politics* knowledge. We set the parameters of this field in Layer 4 to the zero vector and re-evaluate five MMLU domains.
> >
> > | Domain    | Original Score | Ablated Score | Relative Drop |
> > |-----------|----------------|---------------|---------------|
> > | algebra   | 85             | 85            | 0.0%          |
> > | chemistry | 86             | 84            | -2.3%         |
> > | physics   | 92             | 90            | -2.2%         |
> > | politics  | 93             | 75            | -19.4%        |
> > | history   | 95             | 83            | -12.6%        |
> >
> >
> > Table E: Layer-4 knowledge field removal (slot 25).  According to Figure 2 in the main paper, Layer-4 field 25 is strongly associated with *history* and *politics* knowledge. We set the parameters of this field in Layer 4 to the zero vector and re-evaluate five MMLU domains.
> >
> > | Domain    | Original Score | Ablated Score | Relative Drop |
> > |-----------|----------------|---------------|---------------|
> > | algebra   | 85             | 78            | -8.2%         |
> > | chemistry | 86             | 80            | -7.0%         |
> > | physics   | 92             | 89            | -3.3%         |
> > | politics  | 93             | 71            | -23.7%        |
> > | history   | 95             | 69            | -27.4%        |

---

> ### Author Response · Authors · 2025-11-22
> **Responses to Reviewer Yfws [2/3]**
>
> **Theoretical Rationale for Explicit Knowledge Memory.** KMM formalizes *explicit* knowledge storage by separating retrieval variables from the token computation stream. Queries are formed in a compact latent space as (H) (from MLA), while a persistent dictionary ($\mathcal{K}, \mathcal{V}$) encodes domain-level knowledge in addressable fields; retrieval produces a distribution ($\alpha$) over fields and a knowledge contribution mapped back to the model space. This yields three theoretical benefits: (i) **structured specialization**, as the field distribution (\alpha) induces competition among knowledge slots, which in practice aligns with domain-specific activations (see the field–domain heatmaps in Figures 2 and 6); (ii) **transparent interpretability**, since ($\alpha$) provides an explicit provenance signal for which field informed the output; and (iii) **localized editability**, because modifying a single value row in $\mathcal{V}$ affects outputs in proportion to the corresponding mass in $\alpha$, minimizing cross-field interference. Empirically, these properties manifest as domain-specialized fields across layers, supporting the claim that KMM acts as a reusable, explicitly queryable memory complementary to MoE’s computational processing.
>
>
> **Empirical Evidence Supporting the Effectiveness of KMM**. In Figure 4 of the manuscript, adding KMM on top of LFA further speeds convergence beyond LFA alone: at the same target loss, the model with LFA+KMM reaches it in ~7.5k steps vs. 10k for the baseline (1.33× faster), showing that explicit memory yields additional training efficiency gains. These result ground both the rationale (“what” and “why”) and the empirical benefit of storing general knowledge explicitly.
>
> ---
>
> >Q3. Then, there are questionable designs and unclear alignment with motivation. The operation of applying convolutional kernels over split feature groups in hidden states is not well-motivated. This design implicitly assumes that the subcomponents (“bits”) of a hidden vector are independent and separable which is a questionable premise. Furthermore, while the paper motivates the need for local inductive bias across tokens in a sequence, the proposed LFA is applied within the hidden dimensions of each token, which does not clearly align with the stated motivation.
>
>
> **A3.** We believe the reviewer may have misunderstood how the convolution in the Local Fusion Attention (LFA) is applied. The convolution operates **along the sequence dimension** (*i.e.*, across tokens), **not along the hidden feature dimension** within a token.
>
> Formally, as defined in Eq. (3) of the manuscript, $\hat{\mathbf{U}}^{(g)}_t$ can be calculated by
>
> $\sum_{s=0}^{k-1} \mathbf{U}^{(g)}_{t - s + \lfloor k/2 \rfloor} \cdot \Theta^{(g)}[s]$
>
> where $\hat{\mathbf{U}}^{(g)}_t \in \mathbb{R}^{L \times d_h}$ is the hidden-state matrix of group (g), with **$L$** indexing the token positions in the input sequence and **$d_h$** the hidden dimension of that group.
> The index $t-s+\lfloor k/2 \rfloor$ explicitly moves **along the sequence axis ($L$)**, meaning that each output position aggregates information from its $k$ neighboring tokens within the same group.
>
> we have made it more clear in Section 4.2.
>
> ---
>
>
> >Q4. The experimental analysis is also weak. The evaluation is limited in depth. There are no ablation studies or analytical results explaining how LFA and KMM contribute to performance gains.
>
> **A4.** We clarify that our paper already includes controlled ablation experiments (see Section 5.4 in the manuscript) validating the effect of each proposed component.
>
> **Ablation and Analytical Validation**. Sections 5.4 and Figure 4 of the manuscript already present systematic ablations where we add **LFA** and **KMM** step-by-step. The results show clear, interpretable contributions: LFA alone yields a 1.11$\times$ faster convergence with lower perplexity, and adding KMM further improves it to 1.33$\times$ faster convergence and lower validation PPL (31.82 → 29.08). These analyses directly explain how each architectural component contributes to the gains.
>
> **Rationale for Focusing on Pre-training Metrics**. We intentionally evaluate on **pre-training perplexity** rather than downstream tasks, because these metrics most cleanly measure the intrinsic effectiveness of the architecture. Downstream results would require additional instruction-tuning and alignment stages, which are costly and can obscure the contribution of the core model design. By analyzing convergence speed and PPL under identical training settings, we obtain a fair and architecture-focused comparison that isolates the effect of LFA and KMM.

---

> ### Author Response · Authors · 2025-11-22
> **Responses to Reviewer Yfws [3/3]**
>
> >Q5. The claim that AIGCoder achieves superior reasoning and coding performance is particularly unconvincing, since no reasoning reinforcement learning (RL) or specialized training is reported besides pre-training and supervised fine-tuning (SFT). It is unclear how the proposed modules alone could match or outperform models such as Claude or Gemini, which undergo extensive reasoning-specific training.
>
>
> **A5.** Our reasoning and coding gains do **not** rely on reinforcement learning; rather, they arise from (i) architectural improvements that make pre-training more effective, and (ii) a large, cleaned, knowledge-intensive SFT corpus.
>
> * **Architectural improvement → better representations.** Our LFA explicitly fuses adjacent tokens before attention, easing short-range pattern learning while preserving full global attention. KMM provides an explicit, parameterized key–value memory for reusable knowledge access. Together they improve information use at both local and global levels. Empirically, the same validation loss is reached **in 7.5k vs. 10k steps** (1.33$\times$ faster) with lower perplexity under identical settings, indicating stronger representations from pre-training alone.
> * **High-quality SFT data → alignment without RL.** We fine-tune on a curated, large-scale instruction dataset (see Section D.2 of the manuscript for details) spanning diverse, knowledge-intensive tasks, which provides strong supervision without reward models. The paper describes a 10B token SFT data assembled from cleaned public corpora, which supplies high-quanlity data for reasoning/coding without specialized RL.
>
>
> ---
>
> >Q6. Finally, some parts of the paper make exaggerated claims. For example, the authors assert that KMM enables “potentially easier knowledge editing and more interpretable inference,” yet no experiments or explanations are provided to support these statements. Since KMM stores knowledge in a latent parameter space, it is not evident how it improves interpretability or editability in practice. Are there any experiments demonstrating that KMM improves knowledge editing or interpretability?
>
> **A6.** We thank the reviewer for this constructive comment. We acknowledge that empirical results for interpretability and editability are not yet included. We are currently running dedicated experiments and will report the results in next few days.
>
> ---
>
> Q7. Which checkpoints are used for the baselines? Are they base, instruction-tuned, or chat versions?
>
> A7. For **fair comparisons**, all baseline models use their **instruct-tuned versions** as clearly stated in Section 5.1 of our paper.
>
> ---
>
> >Q8. What is the backbone model of AIGCoder?
>
> A8. AIGCoder is a decoder architecture that integrates a DeepSeek MoE backbone with two key innovations: the Local Fusion Attention (LFA) for enhanced local context modeling, and the Knowledge Memory Module (KMM) for explicit global knowledge access.
>
>
> ---
>
> >Q9. Typo: The sentences in lines 174–176 are repeated.
>
> A9. Thank you for pointing out this. We have corrected it and also conducted a thorough proofreading of the  manuscript to eliminate any other typos.
>
> ---
>
> We sincerely hope our clarifications above have addressed your concerns.

---

### Meta-Review · Area_Chair_3HP2 · 2026-01-03

**Summary:**

This paper proposes AIGCoder adding Local Fusion Attention and a Knowledge Memory Module to speed pretraining and improve benchmarks.

**Reviewer Concerns:**

Main concerns from reviewers include:

1. concerns about motivation/justification for LFA/KMM, design–motivation alignment, and missing/unclear technical details (e.g., K/V updates, scaling/parallelism).

2. weak positioning vs prior memory work, possible data contamination from surprising scores, and unsupported/exaggerated claims (interpretability/editability; reasoning gains without RL).

**Reviewer Scores:**

Reviewer / Score

Yfws	2

EeMf	2

jiTz	6

MoAF	2

Average 3

No reviewers indicated to increase or decrease their scores.

---

### Decision · Program_Chairs · 2026-01-26

Reject